# Flattening Sharpness for Dynamic Gradient Projection Memory Benefits Continual Learning

**Danruo Deng**[1], **Guangyong Chen**[2],[*] **Jianye Hao**[3,4], **Qiong Wang**[2], **Pheng-Ann Heng**[1,2]

[1]The Chinese University of Hong Kong, [2]Guangdong-Hong Kong-Macao Joint Laboratory of Human-Machine Intelligence-Synergy Systems, Shenzhen Institute of Advanced Technology, Chinese Academy of Sciences, [3]College of Intelligence and Computing, Tianjin University, [4]Huawei Noah's Ark Lab

{drdeng,pheng}@cse.cuhk.edu.hk, {gy.chen, wangqiong}@siat.ac.cn, jianye.hao@tju.edu.cn

## Abstract

The backpropagation networks are notably susceptible to catastrophic forgetting, where networks tend to forget previously learned skills upon learning new ones. To address such the 'sensitivity-stability' dilemma, most previous efforts have been contributed to minimizing the empirical risk with different parameter regularization terms and episodic memory, but rarely exploring the usages of the *weight loss landscape*. In this paper, we investigate the relationship between the weight loss landscape and sensitivity-stability in the continual learning scenario, based on which, we propose a novel method, *Flattening Sharpness for Dynamic Gradient Projection Memory (FS-DGPM)*. In particular, we introduce a soft weight to represent the importance of each basis representing past tasks in GPM, which can be adaptively learned during the learning process, so that less important bases can be dynamically released to improve the sensitivity of new skill learning. We further introduce *Flattening Sharpness (FS)* to reduce the generalization gap by explicitly regulating the flatness of the weight loss landscape of all seen tasks. As demonstrated empirically, our proposed method consistently outperforms baselines with the superior ability to learn new skills while alleviating forgetting effectively.[2].

## 1 Introduction

Humans have the ability to continually learn new knowledge without forgetting their previously learned ones through mediating a rich set of neurocognitive mechanisms [41, 15, 39]. This ability, often known as continual learning or lifelong learning [29], is crucial for computational systems, such as deep neural networks (DNNs), which are required to sequentially learn and deal with multiple tasks when implemented in the dynamically changing environment. Continual learning remains a long-standing challenge for DNNs since these networks are typically trained with stationary training batches by stochastic gradient descent methods [19], which generally leads to an abrupt performance decrease on previously learned tasks as new tasks are learned. To address such *catastrophic forgetting*, we can brutally retrain an oracle network on the entire dataset containing all tasks to capture dynamic changes in the data distribution, but this methodology is obviously too inefficient to hinder the learning of novel data in real time.

During the last few years, lots of research efforts have been devoted to improving the *stability* of DNNs on old tasks while keeping *sensitive* to new information. The first intuitive idea is to introduce

---

[*]Corresponding author: Guangyong Chen. <gy.chen@siat.ac.cn>

[2]The code is available at: https://github.com/danruod/FS-DGPM

an independent branch for each new task while freezing the old task parameters to preserve the old knowledge [33, 45, 43, 25, 35, 22]. However, in this way, the network will inevitably become redundant as the task number continually increases. As presented in the neurocognitive works [41, 15], the reactivation of neuronal activity patterns, representing old memories, plays an important role in the continual learning of humans [39]. Thus, forgetness can be effectively mitigated by training a single network for new tasks by considering diverse information stored in the memory, including the original training samples of old tasks [31, 5, 13], the gradients induced from old tasks [9] and the feature subspace representing old tasks [34]. However, their continual learning performance is still limited because DNNs can easily overfit the limited information stored in the small-size memory.

The overfitting problem of DNNs is often attributed to the complex loss landscape containing multiple local optima, and the sharpness of the loss landscape has been widely used to characterize the generalization gap in standard training scenarios from both theoretical and empirical perspectives [27, 21, 42, 10, 6]. While this characterization has inspired new approaches for model training with better generalization, practical algorithms that especially seek out flatter minima to effectively address the 'sensitivity-stability' dilemma for continual learning have thus far been elusive. In this paper, our first contribution is to characterize the weight loss landscape for the continual learning scenario and identify that a flatter loss landscape with lower loss value often leads to better continual learning performance, as shown in Figure 1 and Figure 3.

Further, based on our characterization of the weight loss landscape, we find that the recently proposed Gradient Projection Memory (GPM) method [34] maintains the lowest loss value on old tasks among the previously proposed methods by taking gradient steps orthogonal to the subspace representing old tasks. However, its loss landscape on newly learned tasks is the sharpest due to the lack of sufficient subspace left for new task learning. To improve the network's sensitivity, our second contribution is to predict the importance of bases spanning the subspace for old tasks, so that less important bases can be dynamically released. In particular, we introduce a soft weight to indicate the bases importance, which can be dynamically adjusted by combining the *Flattening Sharpness (FS)* to minimize the loss value and loss sharpness simultaneously. Intuitively, a basis will be regarded as important for preserving old knowledge if the gradients induced by new tasks and old ones are aligned in the opposite direction on that basis. As demonstrated through extensive experiments, our proposed method can consistently outperform the state-of-the-art methods [17, 30, 24, 31, 5, 13, 34] by a notable margin across a range of widely used benchmark datasets.

## 2    Related Work

In this section, we briefly survey the representative works of continual learning by highlighting their contributions. To simplify our presentation, this section is organized by dividing the representative works into three categories, parameter isolation, regularization-based, memory-based methods.

**Parameter isolation** methods address forgetting by assigning a different subset of network parameters to each task. Without restrictions on network architecture, new neurons or layers or modules can be added for new tasks, while the previous task parameters can be frozen or copied to preserve old knowledge. For instance, Progressive Neural Network (PGN) [33] freezes the parameters trained with previous knowledge while expands the architecture by allocating new sub-networks with fixed capacity for new tasks. Dynamically Expandable Networks (DEN) [45] selectively retrains or expands network capacity by splitting/duplicating important units on new tasks. Reinforced Continual Learning (RCL) [43] uses reinforcement learning strategy to adaptively expand the network of each layer, while [22] use neural architecture search to find optimal network structures for each sequential task. Alternatively, with the architecture remaining static, a fixed part is allocated to each task. During the training of a new task, previous task parts are masked out to prevent interference. The mask sets are imposed at parameter level [25], or unit level [35]. PackNet [25] uses iterative pruning to fully restrict gradient updates on important weights via a binary mask, whereas HAT [35] limits the update of important units recognized by the hard attention mask through gradient backpropagation.

**Regularization-based** methods introduce an additional regularization term in the loss function to consolidate previous knowledge without using replay. This involves using knowledge distillation [23, 14] or penalizing changes in weights deemed important for previous tasks [17, 46, 28, 2, 3] to reduce forgetting. There are many ways to measure the importance. Elastic Weight Consolidation (EWC) [17] identifies important weights based on the diagonal values of Fisher information matrix

after training, while Synaptic Intelligence (SI) [46] calculates them online and over the entire learning trajectory in parameter space. Memory Aware Synapses (MAS) [2] estimates importance weights in an unsupervised manner, while Variational Continual Learning (VCL) [28] introduces a variational framework that spawned some Bayesian-based works [32, 8, 1, 7]. For example, [32] recursively uses a Gaussian Laplace approximation of the Hessian to approximate the posterior after every task, [8] adjusts the learning rate according to the uncertainty defined by the probability distribution of the network weights. [1] introduces an interpretation of node-wise uncertainty on the Kullback-Leibler (KL) divergence term of the variational lower bound for Gaussian mean-field approximation.

Our method mainly follows **memory-based** methods, which mitigate forgetting based on information extracted from old tasks or based on a generative model to generate pseudo samples. For example, iCaRL [30] selects and stores samples closest to the feature mean of each class. ER [5, 31] suggests reservoir sampling under the limited and fixed budget for replay buffer. Deep Generative Replay (DGR) [36] trains a deep generative model in the Generative Adversarial Network (GAN) framework [11] to simulate past data. These previous task samples are mainly reused as model inputs for replay in the above methods. However, replay might be prone to overfitting the subset of stored samples. Alternatively, samples stored in memory can also be used to constrain the optimization of the new task loss to prevent previous task interference, thereby leaving more leeway for backward and forward transfer. Gradient Episodic Memory (GEM) [24] projects the estimated gradients in the feasible region, which is outlined by previous task gradients calculated from the episodic memory samples. Averaged-GEM (A-GEM) [4] relaxes the projection to a direction that is estimated from samples randomly selected from memory. [12] proposes a unified view of episodic memory-based continual learning methods, including GEM and A-GEM, and improves performance over these methods by using a loss-balancing update scheme. A few other works have utilized gradient information to protect previous knowledge. [31, 13] adopt optimization-based meta-learning to enforce gradient alignment between samples from different tasks. GPM [34] minimizes interference between sequential tasks by ensuring that gradient updates only occur in directions orthogonal to the input of previous tasks.

## 3 The Weight Loss Landscape of Continual Learning

In this section, we first introduce our formulation of continual learning, and then characterize the weight loss landscape for the continual learning scenario from stability and sensitivity. Finally, some insights combining the weight loss landscape and continual learning are provided.

### 3.1 Problem Formulation

Throughout the paper, we denote scalars as $a$, vectors as $\boldsymbol{a}$, matrices as $\boldsymbol{A}$, and sets as $\mathcal{A}$. We consider a supervised learning setup where $T$ tasks are sequentially learned from their training data. Each task has an identical task descriptor, $\tau \in \{1, 2, \ldots, T\}$, with its dataset $\mathcal{D}_\tau = \{\boldsymbol{x}_{i,\tau}, y_{i,\tau}\}_{i=1}^{n_\tau}$ containing $n_\tau$ samples randomly generated from a latent distribution $\mathscr{D}_\tau$. At any time-step during the learning process, we minimize the empirical risk of the model on all $t$ tasks seen so far, with just limited size of memory $\mathcal{M}$ to summarize the training data of previous tasks $\{\mathcal{D}_\tau\}_{\tau=1}^{t-1}$. To simplify the notation, we denote $L_\mathcal{A}(\boldsymbol{w}) = \frac{1}{|\mathcal{A}|} \sum_{(\boldsymbol{x}, y) \in \mathcal{A}} [\ell(f_{\boldsymbol{w}}(\boldsymbol{x}), y)]$ as the average empirical loss for the set $\mathcal{A}$, where $\ell(\cdot, \cdot)$ is an arbitrary loss function (*e.g.* the cross-entropy (CE) loss), $|\mathcal{A}|$ is the sample size of the set $\mathcal{A}$, and $f_{\boldsymbol{w}}$ is the DNN with weight $\boldsymbol{w}$. Our final goal is to find an optimal parameter $\boldsymbol{w}$, which minimizes the overall risk $\sum_{\tau=1}^{T} L_{\mathscr{D}_\tau}(\boldsymbol{w})$ for all tasks.

### 3.2 Connection of Weight Loss Landscape and Continual Learning

After learning a new task, we visualize the weight loss landscape of each task seen so far by plotting changes in its training loss when moving the weights $\boldsymbol{w}$ in a random direction $\boldsymbol{d}$ with magnitude $\alpha$ following [21]:

$$g_t(\alpha) = L_{\mathcal{D}_t}(\boldsymbol{w} + \alpha\boldsymbol{d}) = \frac{1}{|\mathcal{D}_t|} \sum_{(\boldsymbol{x}, y) \in \mathcal{D}_t} \ell(f_{\boldsymbol{w}+\alpha\boldsymbol{d}}(\boldsymbol{x}), y),$$

where $\mathcal{D}_t$ is the training set for the $t$-th task previously learned. To eliminate the scaling invariance of DNNs, $\boldsymbol{d}$ is sampled from a Gaussian distribution and further normalized by $\boldsymbol{d}_{l,j} \leftarrow \frac{\boldsymbol{d}_{l,j}}{\|\boldsymbol{d}_{l,j}\|_F} \|\boldsymbol{w}_{l,j}\|_F$,

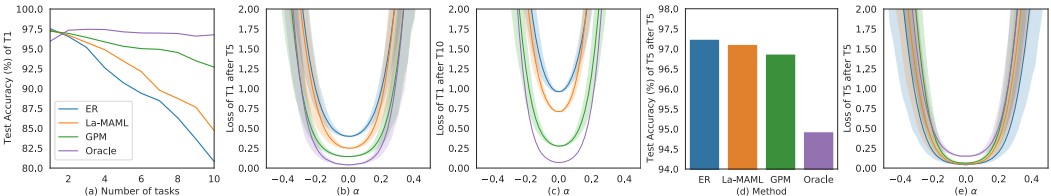

Figure 1: The connection between the weight loss landscape and continual learning is investigated on four methods. (a)-( c) shows the stability of the first task. (a) is the test accuracy change curve of the first task; (b) and (c) are the weight loss landscape of the first task after learning the fifth task and all ten tasks. (d)-(e) shows the sensitivity of the fifth task, which are the test accuracy and the weight loss landscape of the fifth task after just learning the fifth task. The shape of the weight loss landscape obtained using ten different random filter-normalized directions. ("T$i$" is abbr. of the $i$-th task)

where $d_{l,j}$ represents the $j$-th filter at $l$-th layer of $d$, and $\|\cdot\|_F$ denotes the Frobenius norm. Compared with our visualization, [26] only consider one task and plot the loss landscape along the directions of its Hessian eigenvectors, which only reflects some of the relationship between forgetting and landscape. Considering $d$ is randomly selected, we repeat the visualization 10 times with different $d$.

We first study the *stability* of the network by plotting changes of the weight loss landscape for the first task after new task learning. In particular, We use the previously proposed ER [5], La-MAML [13], and GPM [34] to train a MLP network with two hidden layers on the Permuted MNIST (PMNIST) [20] dataset that contains 10 tasks. We also retrain the network on the entire dataset contain all passed tasks as an Oracle network. Early stopping is used to halt the training with up to 10 epochs for each task based on the validation loss as proposed in [35]. As shown in Figure 1(a), all three continual learning methods lose their stability as learning new tasks. It can be observed from Figure 1(b)-(c) that the weight loss landscape becomes sharper and loss value increases simultaneously, when the testing accuracy of the first task continually decreases. We further evaluate the *sensitivity* of the network by observing the performance of the fifth task just after it has just been learned. As shown in Figure 1(d)-(e), ER shows the best learning capability compared with other methods with the lowest loss value and the flattest loss landscape. Thus, based on these empirical findings, we assume that lower loss value with a flatter neighbor may lead to better continual learning performance.

### 3.3 A Case Study of Flattening Sharpness for Vanilla ER

In this part, we further validate our above assumption by Flattening Sharpness for vanilla ER (FS-ER). Compared with ER that looks for a solution $w$ that jointly minimizes the training loss of current task data and memory data, FS-ER seeks out a solution with both low loss and flat neighbor by minimizing the maximal loss in the neighbor around the parameter value. The schematic of the Flattening Sharpness (FS) is shown in Figure 2. We introduce the adversarial weight perturbation (orange dashed line) to explicitly flatten the weight loss landscape via injecting the worst-case weight perturbation, which is calculated from the current task data and past task data sampled from the replay buffer (Refer to Appendix C.1 and the next section for more details). Figure 3 respectively shows the weight loss landscapes of the fifth task after just learning the fifth task and all tasks. We find that FS-ER successfully gets a solution with a lower loss value and flatter landscape, either after the fifth task has just been learned or all ten tasks have been learned. The average testing accuracy of all tasks using FS-ER is **90.44%**, significantly higher than ER (**86.16%**), which means that flattening sharpness does benefit continual learning.

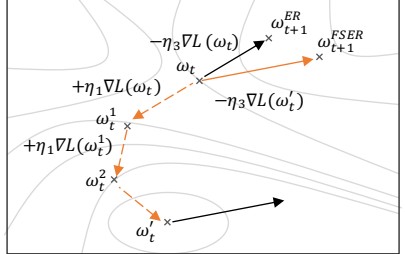

Figure 2: Schematic of FS-ER update. The dashed line and the solid line indicate the gradient ascent and descent, respectively. Orange denotes the actual update of the parameter $w$.

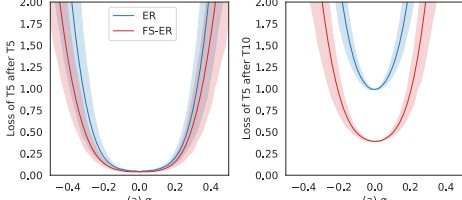

Figure 3: Landscape of the fifth task after just learning (a) the fifth task and (b) all ten tasks.

# 4 Flattening Sharpness for Dynamic Gradient Projection Memory

As shown in Figure 1, GPM achieves the highest testing accuracy on old tasks among all three practical continual learning methods, but shows less sensitivity to new task learning. To address this issue, we propose *Flattening Sharpness for Dynamic GPM (FS-DGPM)*, which dynamically adjusts the gradient subspace representing the past tasks to improve the sensitivity to the new task, while ensuring stability of the previous tasks. In particular, we let $M = [u_1, u_2, \cdots, u_k]$ denote the bases matrix that spans the gradient subspace of the previous task, $\Lambda = \text{diag}[\lambda_1, \lambda_2, ..., \lambda_k]$ be the diagonal matrix with its $i$-th diagonal element $\lambda_i \in [0, 1]$ indicating the importance of each basis, and $k$ is the number of bases.

## 4.1 Sharpness Evaluation

Comparing with the classical strategy that perturbs weight in the entire space [40, 16, 42, 10], we focus on characterizing the weight loss landscape on the new task with respect to the important subspace representing old task. The important subspace can be effectively calculated based on the examples sampled from replay buffer $\mathcal{M}$ after each task training. Then, we can find the worst case by maximizing the training loss of the network on the new task in this subspace. Formally, the sharpness of the loss landscape around the solution $w$ in the old parameter space can be predicted as,

$$\max_{v \in \mathcal{V}} \quad L_{\hat{\mathcal{D}}_t}(w + v), \tag{1}$$

where $\mathcal{V}$ denotes the subspace spanned by $M$ and $\Lambda$, and $\hat{\mathcal{D}}_t$ denotes the batch samples of the current $t$-th task. As shown in Eq. (1), the high value can be obtained when the network fails to learn the new task (*sensitivity*) and the new task learning seriously interferes with the past tasks learning (*stability*). Based on the gradient method, the adversarial weight perturbation $v$ can be solved as,

$$v \leftarrow v + \eta_1 M \Lambda M^T \left( \nabla_{(w+v)} L_{\hat{\mathcal{D}}_t}(w + v) \right), \tag{2}$$

where $\eta_1$ is the update step size. Note that $v$ is initialized as $\mathbf{0}$ and layer-wise updated. As shown in Appendix E, two-step for $v$ (default settings) are enough to get good improvements.

## 4.2 Dynamic Gradient Projection Memory

After obtaining the adversarial weight perturbation $v$, we can further update the bases importance matrix $\Lambda = \text{diag}[\lambda_1, \lambda_2, ..., \lambda_k]$ by jointly considering the current task batch $\hat{\mathcal{D}}_t$ and the batch $\hat{\mathcal{M}}$ sampled from the replay buffer $\mathcal{M}$ as following,

$$\lambda_i \leftarrow \lambda_i - \eta_2 \left( \nabla_{\lambda_i} L_{\hat{\mathcal{D}}_t \cup \hat{\mathcal{M}}}(w + v) \right), \tag{3}$$

where the sigmoid function is used at the end of gradient update to constrain the importance value $\lambda_i$ between 0 and 1. In addition, the second term on the right side in Eq. (3) can be approximated by the first-order Taylor expansion as,

$$\nabla_{\lambda_i} L_{\hat{\mathcal{D}}_t \cup \hat{\mathcal{M}}}(w + v) \approx \eta_1 \left( \nabla_w L_{\hat{\mathcal{D}}_t}(w) \right)^T u_i u_i^T \left( \nabla_w L_{\hat{\mathcal{D}}_t \cup \hat{\mathcal{M}}}(w) \right). \tag{4}$$

The above equation characterizes the relationship between the gradients induced by the current task and the old tasks based on the basis $u_i$. As illustrated in the Figure 4, this equation implies that when the projections of two gradients on the basis $u_i$ are aligned in the same direction, the gradient of $\lambda_i$ will be positive, and when there is interference, the gradient will be negative. The positive (negative) gradient will decrease (increase) the importance $\lambda_i$, thereby releasing (tightening) the update limit of the new task on the corresponding basis $u_i$. We provide the full derivation in the Appendix A.2. Note that the initial value of all importance is set to 1 and dynamically adjusted from the second task.

(a) Transfer  (b) Interference

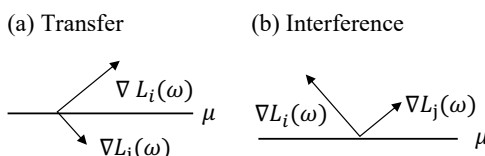

Figure 4: A depiction of transfer (a) and interference (b) in the basis $u$ of gradient space.

## 4.3 Weight Updating

Finally, we update the model parameters by minimizing the worst performance of $f_{\boldsymbol{w}+\boldsymbol{v}}$ with respect to $\boldsymbol{w}$, while adjusting the update magnitude of $\boldsymbol{w}$ on each basis based on its importance to alleviate forgetting. More concretely, the parameter $\boldsymbol{w}$ will be updated to:

$$\boldsymbol{w} \leftarrow \boldsymbol{w} - \eta_3 \left( \boldsymbol{I} - \boldsymbol{M}\boldsymbol{\Lambda}\boldsymbol{M}^T \right) \nabla_{\boldsymbol{w}} L_{\hat{\mathcal{D}}_t \cup \hat{\mathcal{M}}}(\boldsymbol{w} + \boldsymbol{v}). \tag{5}$$

Note that the optimization is performed over the model parameters $\boldsymbol{w}$, whereas the objective is computed using perturbed model $f_{\boldsymbol{w}+\boldsymbol{v}}$. In addition, we update the replay buffer $\mathcal{M}$ with reservoir sampling as in [31], and then use Singular Value Decomposition (SVD) to recalculate $\boldsymbol{M}$ based on the sampling data in the replay buffer after learning one task following GPM [34]. Comparing with [34], we calculate the important bases in the entire gradient space and use them to replace the bases calculated last time. Besides that, our method degenerates to GPM when $\eta_1$ and $\eta_2$ are set to 0. The complete pseudo-code of FS-DGPM is outlined in the Algorithm 1.

---

**Algorithm 1** FS-DGPM (Flattening Sharpness for Dynamic Gradient Projection Memory)

---

**Input:** Network weight $\boldsymbol{w}$, loss function $\ell$, learning rate $\eta_3$, FS step size $\eta_1$, FS steps $K$, Soft weight step size $\eta_2$, batch size $b$.
Initializing $\mathcal{M} \leftarrow \{\}, \boldsymbol{M} \leftarrow \boldsymbol{I}, \boldsymbol{\Lambda} \leftarrow \boldsymbol{I}$
**for** $t = 1, 2, \cdots, T$ **do**
    **for** $ep = 1, 2, \cdots, num_{epochs}$ **do**
        **for** batch $\hat{\mathcal{D}}_t \overset{b}{\sim} \mathcal{D}_t$ **do**
            $\hat{\mathcal{M}} \overset{b}{\sim} \mathcal{M}$
            **for** $k = 1, \cdots, K$ **do**
                $\boldsymbol{v} \leftarrow \boldsymbol{v} + \eta_1 \boldsymbol{M}\boldsymbol{\Lambda}\boldsymbol{M}^T \left( \nabla_{(\boldsymbol{w}+\boldsymbol{v})} L_{\hat{\mathcal{D}}}(\boldsymbol{w} + \boldsymbol{v}) \right)$       ▷ Sharpness Evaluation
            **end for**
            **if** $t \geq 2$ **then**
                $\boldsymbol{\Lambda} \leftarrow \boldsymbol{\Lambda} - \eta_2 \left( \nabla_{\boldsymbol{\Lambda}} L_{\hat{\mathcal{D}}_t \cup \hat{\mathcal{M}}}(\boldsymbol{w} + \boldsymbol{v}) \right)$     ▷ Dynamic Gradient Projection Memory
            **end if**
            $\boldsymbol{w} \leftarrow \boldsymbol{w} - \eta_3 \left( \boldsymbol{I} - \boldsymbol{M}\boldsymbol{\Lambda}\boldsymbol{M}^T \right) \nabla_{\boldsymbol{w}} L_{\hat{\mathcal{D}}_t \cup \hat{\mathcal{M}}}(\boldsymbol{w} + \boldsymbol{v})$       ▷ Weight updating
            Push $\hat{\mathcal{D}}_t$ to $\mathcal{M}$ with reservior sampling
        **end for**
    **end for**
    $\boldsymbol{M} \leftarrow \text{UpdateGPM}(\mathcal{M})$             ▷ see Appendix Alg. 2
**end for**

---

## 4.4 Theoretical Understanding

We further provide a theoretical view on why landscape can characterize the continual learning performance and why our proposed FS-DGPM works. To simplify our explanation, we only consider two tasks, which contains the training sets $\mathcal{D}_1$ and $\mathcal{D}_2$ sampled from the distributions $\mathscr{D}_1$ and $\mathscr{D}_2$, respectively. Based on the previous works on PAC-Bayes bound [27, 42, 10], given a "prior" distribution $P$ (a common assumption is zero mean, $\sigma^2$ variance Gaussian distribution) over the weights, the expected error of the classifier for the continual learning scenario can be bounded with probability at least $1 - \delta$ over the draw of $n$ training data:

$$\min_{\Delta\boldsymbol{w}} \mathbb{E}_{\boldsymbol{v}} \left[ L_{\mathscr{D}_1 \cup \mathscr{D}_2}(\boldsymbol{w} + \Delta\boldsymbol{w} + \boldsymbol{v}) \right] \leq \min_{\Delta\boldsymbol{w} \in \mathcal{V}^C} \mathbb{E}_{\boldsymbol{v}} \left[ L_{\mathscr{D}_1}(\boldsymbol{w} + \Delta\boldsymbol{w} + \boldsymbol{v}) \right] + L_{\mathcal{D}_2}(\boldsymbol{w} + \Delta\boldsymbol{w})$$

$$+ \max_{\boldsymbol{v} \in \mathcal{V}} L_{\mathcal{D}_2}(\boldsymbol{w} + \Delta\boldsymbol{w} + \boldsymbol{v}) - L_{\mathcal{D}_2}(\boldsymbol{w} + \Delta\boldsymbol{w}) + 4\sqrt{\frac{1}{n} \left( KL(\boldsymbol{w} + \Delta\boldsymbol{w} + \boldsymbol{v}||P) + \ln\frac{2n}{\delta} \right)}.$$

where $\Delta\boldsymbol{w}$ is the update based on the previously optimal solution $\boldsymbol{w}$ learned on the old task $\mathcal{D}_1$ when learning the new one $\mathcal{D}_2$, and $\boldsymbol{v}$ is often chosen as a zero mean spherical Gaussian perturbation with variance $\sigma^2$ in every direction. Let $\Delta\boldsymbol{w} \in \mathcal{V}^C$, then $\Delta\boldsymbol{w}$ lies in the complementary space of the important space representing the old task $\mathscr{D}_1$, so that $\mathbb{E}_{\boldsymbol{v}} \left[ L_{\mathscr{D}_1}(\boldsymbol{w} + \Delta\boldsymbol{w} + \boldsymbol{v}) \right]$ does not increase

too much compared with the previously minimized $\mathbb{E}_{\boldsymbol{v}}\left[L_{\mathcal{D}_1}(\boldsymbol{w}+\boldsymbol{v})\right]$. The second term denotes the empirical loss on the second task and the third term represents the sharpness of the weight loss landscape around the $\boldsymbol{w}+\Delta\boldsymbol{w}$. Since we have constrained $\Delta\boldsymbol{w} \in \mathcal{V}^C$, then it is natural to assume $\boldsymbol{v} \in \mathcal{V}$, so that $\Delta\boldsymbol{w}+\boldsymbol{v}$ will cover the full space. Thus, our FS-DGPM exactly optimizes the worst-case of the flatness of weight loss landscape to control the PAC-Bayes bound, which theoretically justifies both lower loss value and flatter landscape lead to better continual learning performance, and why our proposed FS-DGPM works.

# 5 Experiments

In this section, we conduct extensive experiments to compare the performance of our proposed FS-DGPM model with the state-of-the-art methods on widely used continual learning benchmark datasets. Additional results and more details about the datasets, experiment setup, baselines, and model architectures are presented in the Appendix D and E.

## 5.1 Experimental Setup

**Datasets:**  We evaluate our algorithm on four image classification datasets: **Permuted MNIST (PMNIST)** [20], **CIFAR-100 Split** [18], **CIFAR-100 Superclass** [44] and **TinyImageNet** [37]. The PMNIST dataset is a variant of the MNIST dataset, in which each task applies a fixed random pixel permutation to the original dataset. The PMNIST benchmark dataset consists of 20 tasks, and each contains only 1000 samples from 10 different classes [13]. The CIFAR-100 Split is constructed by randomly dividing 100 classes of CIFAR-100 into 10 tasks with 10 classes per task. The CIFAR-100 Superclass is divided into 20 tasks according to the 20 superclasses of the CIFAR-100 dataset, and each superclass contains 5 different but semantically related classes. Whereas, TinyImageNet is constructed by splitting 200 classes into 40 5-way classification tasks. In our experiments, we do not use any data augmentation. The dataset statistics are given in Appendix D.1.

**Network Architecture:**  For PMNIST, we use a fully connected network with two hidden layers of 100 units each following [24]. For experiments of CIFAR-100 Split and CIFAR-100 Superclass, we use a 5-layer AlexNet and LeNet architecture similar to [34] respectively. For TinyImageNet, we use the same network architecture as [13], which consists of 4 conv layers and 3 fully connected layers. In PMNIST, all tasks share the final classifier layer, while other experiments use a multi-head incremental setting, that is, each task has a separate classifier. Refer to Appendix D.2 for more details.

**Baselines:**  We compare our method against multiple methods described below. (1) **EWC** [17], a regularization-based method that uses the diagonal of Fisher information to identify important weights; (2) **ICARL** [30], a memory-based method that uses knowledge-distillation and episodic memory to reduce forgetting; (3) **GEM** [24], another memory-based method that uses the gradient of episodic memory to constrain optimization to prevent forgetting; (4) **ER** [5], a simple and competitive method based on reservoir sampling; (5) **La-MAML** [13] and (6) **GPM** [34] are memory-based methods inspired by optimization-based meta-learning and based on gradient orthogonal constraints, respectively; (7) **Multitask** is an oracle baseline that all tasks are learned jointly using the entire dataset at once in a single network. Multitask is not a continual learning strategy but serves as an upper bound on average test accuracy on all tasks.

**Training Details:**  All baselines and our method use stochastic gradient descent (SGD) for training. For each task in PMNIST and TinyImageNet, we train the network in 1 and 10 epochs, respectively, with the batch size as 10. These experimental settings are the same as La-MAML [13], so that we directly compare with their reported results. In the CIFAR-100 Split and CIFAR-100 Superclass experiments, we use the early termination strategy to train up to 50 epochs for each task, which is based on the validation loss as proposed in [35]. For both datasets, the batch size is set to 64. The replay buffer size of PMNIST, CIFAR-100 Split, CIFAR-100 Superclass, and TinyImageNet are 200, 1000, 1000, and 400, respectively. Details about the experimental setting and the hyperparameters considered for each baseline are provided in Appendix D.5 and D.6.

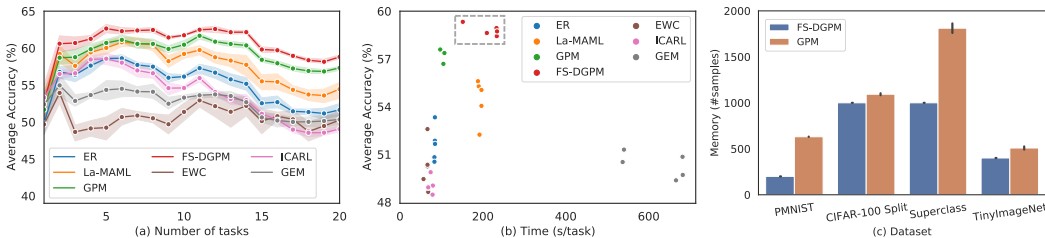

Figure 5: (a) Average accuracy as a function of the number of tasks trained on 20-Split CIFAR-100 Superclass. (b) Training time per task on 20-Split CIFAR-100 Superclass. (c) Memory usage on four datasets. ("Superclass" is abbr. of CIFAR-100 Superclass).

Table 1: Experimental results on 10-Split CIFAR-100, 20-Split CIFAR-100 Superclass and 40-Split TinyImageNet in 50 epochs. Each experiment is run with 5 seeds. $^\dagger$ and $^*$ denotes results reported by [13] and [44] respectively.

| Method | CIFAR-100 Split | | CIFAR-100 Superclass | | TinyImageNet | |
|---|---|---|---|---|---|---|
| | ACC(%) | BWT(%) | ACC(%) | BWT(%) | ACC(%) | BWT(%) |
| EWC | $72.77 \pm 0.45$ | $-3.59 \pm 0.55$ | $50.26 \pm 1.48$ | $-7.87 \pm 1.63$ | - | - |
| GEM | $70.15 \pm 0.34$ | $-8.61 \pm 0.42$ | $50.35 \pm 0.80$ | $-9.50 \pm 0.85$ | $50.57 \pm 0.61^\dagger$ | $-20.50 \pm 0.10^\dagger$ |
| ICARL | $53.50 \pm 0.81$ | $-20.44 \pm 0.82$ | $49.05 \pm 0.51$ | $-11.24 \pm 0.27$ | $54.77 \pm 0.32^\dagger$ | $-3.93 \pm 0.55^\dagger$ |
| ER | $70.07 \pm 0.35$ | $-7.70 \pm 0.59$ | $51.64 \pm 1.09$ | $-7.86 \pm 0.89$ | $48.32 \pm 1.51^\dagger$ | $-19.86 \pm 0.70^\dagger$ |
| La-MAML | $71.37 \pm 0.67$ | $-5.39 \pm 0.53$ | $54.44 \pm 1.36$ | $-6.65 \pm 0.85$ | $66.90 \pm 1.65^\dagger$ | $-9.13 \pm 0.90^\dagger$ |
| GPM | $73.18 \pm 0.52$ | $\mathbf{-1.17 \pm 0.27}$ | $57.33 \pm 0.37$ | $\mathbf{-0.37 \pm 0.12}$ | $67.39 \pm 0.47$ | $\mathbf{1.45 \pm 0.22}$ |
| **FS-DGPM** | $\mathbf{74.33 \pm 0.31}$ | $-2.71 \pm 0.17$ | $\mathbf{58.81 \pm 0.34}$ | $-2.97 \pm 0.35$ | $\mathbf{70.41 \pm 1.30}$ | $-2.11 \pm 0.84$ |
| Multitask | $79.75 \pm 0.38$ | - | $61.00 \pm 0.20^*$ | - | $77.10 \pm 1.06^\dagger$ | - |

**Metrics:** We evaluate the continual learning performance by the *average accuracy* (ACC) and *backward transfer* (BWT) [24, 4, 5], formulated as following,

$$ACC = \frac{1}{T}\sum_{i=1}^{T} R_{T,i}, \quad BWT = \frac{1}{T-1}\sum_{i=1}^{T-1} R_{T,i} - R_{i,i},$$

where $T$ is the total number of learned sequential tasks, $R_{i,j}$ is the test classification accuracy of the model on $j$-th task after learning the last sample from $i$-th task. ACC is the average test classification accuracy of all tasks, bigger is better. BWT is the interference of new learning on the past knowledge. More specifically, negative BWT implies (catastrophic) forgetting whereas positive BWT indicates learning new task increases the performance on some preceding tasks.

## 5.2 Results and Discussion

**PMNIST:** We first evaluate our algorithm for 20 sequential PMNIST tasks with only 1000 samples per task in a single-head setting. From the results, as shown in Table 2, we see that our method (FS-DGPM) achieves the best average accuracy (**76.96% $\pm$ 0.77**). Moreover, we achieve the least amount of forgetting except GPM, which is essentially a trade-off in accuracy to minimize forgetting. As shown in Figure 5(c), we only use about **31%** of the final memory size of GPM and achieve $\sim$ **2.5%** better accuracy.

Table 2: Experimental results (mean $\pm$ std in 5 runs) on PMNIST in single-epoch.

| Method | PMNIST | |
|---|---|---|
| | ACC(%) | BWT(%) |
| EWC | $62.25 \pm 1.44$ | $-15.22 \pm 1.25$ |
| GEM | $61.82 \pm 0.85$ | $-15.58 \pm 1.17$ |
| ER | $68.31 \pm 0.56$ | $-13.91 \pm 0.67$ |
| La-MAML | $75.98 \pm 0.60$ | $-10.21 \pm 0.90$ |
| GPM | $74.54 \pm 0.36$ | $\mathbf{-7.17 \pm 0.51}$ |
| **FS-DGPM** | $\mathbf{76.96 \pm 0.51}$ | $-7.45 \pm 0.30$ |
| Multitask | $86.54 \pm 1.74$ | - |

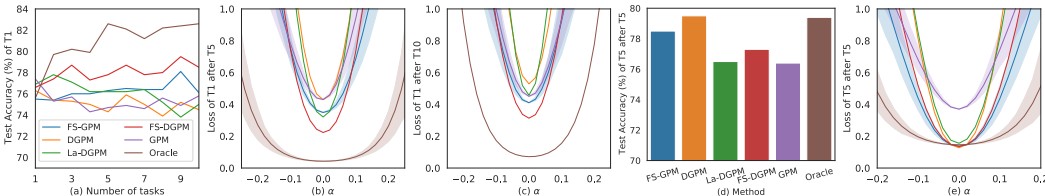

Figure 6: The ablation study implemented on CIFAR-100 Split with 10 tasks. (a)-(c) shows the stability of the first task. (a) is the test accuracy change curve of the first task; (b) and (c) are the weight loss landscape of the first task after learning the fifth task and all ten tasks. (d)-(e) shows the sensitivity of the fifth task, which are the test accuracy and the weight loss landscape of the fifth task after just learning the fifth task. The shape of the weight loss landscape obtained using ten different random filter-normalized directions. ("T$i$" is abbr. of the $i$-th task)

**CIFAR-100 and TinyImageNet:** Next, we use a multi-head setting to evaluate our algorithm under the more challenging visual classification benchmarks. Table 1 reports all results of these experiments. We outperform all baselines on three datasets, with achieving the best average accuracy **74.33%**, **58.81%** and **70.41%**. In these experiments, we observe that GPM is a strong baseline with the least forgetting. At the same time, we highlight that our method achieves the highest accuracy on all datasets and the second-lowest forgetting after GPM. Figure 5(a) shows the process of performance changing with the number of tasks on the CIFAR-100 Superclass. We consistently see the superior performance of our method at any stage of model evolution. It is also worth emphasizing that although our method requires more time for training than GPM, it has lower memory usage and better test accuracy (See Figure 5(b)-(c)). As noted by [4] and [38], EWC performs poorly without multiple passes over the datasets, and GEM is not very effective under the single-head variants. These situations have also been observed in our experiments.

## 5.3 Ablation studies on FS-DGPM

We further investigate our model performance with an ablation study and summarize it in Table 3. We respectively ablate the effects of flattening sharpness and dynamically adjusting the soft weight for bases. We refer to them as DGPM and FS-GPM. We also construct an ablation referred to as La-DGPM (Look-ahead DGPM), where the adversarial weight perturbation $v$ in Eq. (2) is changed to the direction

Table 3: The ablation study results on CIFAR-100 Split and Superclass. Each experiment is run with 5 seeds.

| | **CIFAR-100 Split** | | **CIFAR-100 Superclass** | |
|---|---|---|---|---|
| **Method** | ACC(%) | BWT(%) | ACC(%) | BWT(%) |
| FS-DGPM | **74.33** $\pm$ 0.31 | **-2.71** $\pm$ 0.17 | **58.81** $\pm$ 0.34 | -2.97 $\pm$ 0.35 |
| La-DGPM | 73.74 $\pm$ 0.61 | -3.05 $\pm$ 0.73 | 58.18 $\pm$ 0.41 | **-2.41** $\pm$ 0.39 |
| FS-GPM | 73.96 $\pm$ 0.44 | -3.12 $\pm$ 0.43 | 58.61 $\pm$ 0.53 | -2.79 $\pm$ 0.30 |
| DGPM | 73.78 $\pm$ 0.32 | -3.67 $\pm$ 0.42 | 56.78 $\pm$ 0.49 | -2.44 $\pm$ 0.40 |
| GPM | 73.18 $\pm$ 0.52 | **-1.17** $\pm$ 0.27 | 57.33 $\pm$ 0.37 | **-0.37** $\pm$ 0.12 |

of gradient descent. At the same time, we also change the sign in Eq. (3) to ensure that the soft weight of the basis is reduced when the gradients are aligned in the same direction. From the results, shown in Table 3, we observe that flattening sharpness does benefits GPM, with $\sim$ **1.0%** improvement over GPM on both datasets. We can further observe through Figure 6 that all landscapes of FS-DGPM have lower loss values and flatter neighbors than DGPM and La-DGPM on the CIFAR-100 Split experiments. In addition, we see that DGPM performs well in learning new tasks, but it also leads to forgetting previous tasks. This situation can be efficiently alleviated by flattening sharpness. Hence, FS is indeed a powerful method worthy of being widely adopted for continual learning scenarios.

## 6 Conclusion

In this paper, we explore the weight loss landscape to characterize the well-known 'sensitivity-stability' dilemma faced by continual learning algorithms, and find that lower loss value with flatter neighbor often leads to better continual learning performance. Based on this finding, we propose our FS-DGPM algorithm, which introduces a soft weight to represent the importance of each basis representing past tasks in GPM, so that less important bases can be dynamically released to improve

the sensitivity of new skill learning. Flattening Sharpness (FS) is also introduced here to reduce the generalization gap by explicitly regulating the flatness of the weight loss landscape of all tasks seen so far. The evaluation of various image classification tasks with different network architectures and the comparison with some state-of-the-art algorithms show the effectiveness of our method in achieving high classification performance while alleviating forgetting. Although our method theoretically and empirically demonstrates the advantages of introducing FS into continual learning, whether there exists a closer upper bound for the continual learning performance remains an unresolved problem and left for our future exploration. Although our method theoretically and empirically demonstrates the advantages of introducing bases soft weight and FS into continual learning, whether there exists a better dynamic adjustment and a closer upper bound for the continual learning performance remains an unresolved problem and left for our future exploration.

## Acknowledgments

We would like to thank Dr. Yongzhe Deng and Zhuo Zhang for their helpful discussions, and anonymous reviewers for their valuable comments to improve this work. This work was supported by a grant from the National Key Research and Development Program of China (Project No. 2020YFB1313900), Hong Kong Research Grants Council under General Research Fund (Project No. 14201620), the National Natural Science Foundation of China (Project No. 62006219, 62072452) and Guangdong Provincial Basic and Applied Basic Research Fund-Regional Joint Fund (Project No. 2020B1515130004).

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
