# A Appendix

## A.1 PAC Bayesian Bound

In this part, we provide a detailed PAC-Bound based on the continual learning scenario.

Based on the previous works on PAC-Bayes bound [27, 42, 10], let $\ell(\cdot, \cdot)$ be 0-1 loss, then for the empirical loss over the training set $\mathcal{D} \sim \mathscr{D}$, we have $L_\mathcal{D}(\boldsymbol{w}) = \frac{1}{|\mathcal{D}|} \sum_{(\boldsymbol{x}, y) \in \mathcal{D}} [\ell(f_{\boldsymbol{w}}(\boldsymbol{x}), y)] \in [0, 1]$. Given a "prior" distribution $P$ (a common assumption is zero mean, $\sigma^2$ variance Gaussian distribution) over the weights of the form $\boldsymbol{w} + \boldsymbol{v}$, the expected error of $f_{\boldsymbol{w}+\boldsymbol{v}}$ can be bounded with probability at least $1 - \delta$ over the draw of $n$ training data:

$$\mathbb{E}_{\boldsymbol{v}} [L_\mathscr{D}(\boldsymbol{w} + \boldsymbol{v})] \leq L_\mathcal{D}(\boldsymbol{w}) + \mathbb{E}_{\boldsymbol{v}} [L_\mathcal{D}(\boldsymbol{w} + \boldsymbol{v})] - L_\mathcal{D}(\boldsymbol{w}) + 4\sqrt{\frac{1}{n} \left( KL(\boldsymbol{w} + \boldsymbol{v} || P) + \ln \frac{2n}{\delta} \right)}, \tag{6}$$

where $\boldsymbol{w}$ is the weight of the predictor learned from the training set, $\boldsymbol{v}$ is a random variable and is often chosen as a zero mean spherical Gaussian perturbation with variance $\sigma^2$ in every direction.

We now consider the bound in the continual learning scenario. To simplify our explanation, we only consider two tasks, which contains the training sets $\mathcal{D}_1$ and $\mathcal{D}_2$ sampled from the distributions $\mathscr{D}_1$ and $\mathscr{D}_2$, respectively. We assume the model $f_{\boldsymbol{w}}$ is learned from the training set $\mathcal{D}_1$ ($\boldsymbol{w} = \text{argmin}_{\boldsymbol{w}} L_{\mathcal{D}_1}(\boldsymbol{w})$, and then continue to learn the training set $\mathcal{D}_2$. Our final goal is to find an optimal parameter $\Delta \boldsymbol{w}$ to minimizes the overall risk $L_{\mathscr{D}_1 \cup \mathscr{D}_2}(\boldsymbol{w} + \Delta \boldsymbol{w})$ for all tasks as follows:

$$\min_{\Delta \boldsymbol{w}} L_{\mathscr{D}_1 \cup \mathscr{D}_2}(\boldsymbol{w} + \Delta \boldsymbol{w})$$

Based on Eq. (6), the expected error of $f_{\boldsymbol{w} + \Delta \boldsymbol{w} + \boldsymbol{v}}$ can be bounded with probability at least $1 - \delta$ over the draw of training set $\mathcal{D}_2$:

$$\mathbb{E}_{\boldsymbol{v}} [L_{\mathscr{D}_1 \cup \mathscr{D}_2}(\boldsymbol{w} + \Delta \boldsymbol{w} + \boldsymbol{v})] = \mathbb{E}_{\boldsymbol{v}} [L_{\mathscr{D}_1}(\boldsymbol{w} + \Delta \boldsymbol{w} + \boldsymbol{v})] + \mathbb{E}_{\boldsymbol{v}} [L_{\mathscr{D}_2}(\boldsymbol{w} + \Delta \boldsymbol{w} + \boldsymbol{v})]$$
$$\leq \underbrace{\mathbb{E}_{\boldsymbol{v}} [L_{\mathscr{D}_1}(\boldsymbol{w} + \Delta \boldsymbol{w} + \boldsymbol{v})]}_{\text{stability of old task}} + \underbrace{L_{\mathcal{D}_2}(\boldsymbol{w} + \Delta \boldsymbol{w})}_{\text{sensitivity of new task}} + \underbrace{\mathbb{E}_{\boldsymbol{v}} [L_{\mathcal{D}_2}(\boldsymbol{w} + \Delta \boldsymbol{w} + \boldsymbol{v})] - L_{\mathcal{D}_2}(\boldsymbol{w} + \Delta \boldsymbol{w})}_{\text{expected sharpness on training set of new task}}$$
$$+ 4\sqrt{\frac{1}{n} \left( KL(\boldsymbol{w} + \Delta \boldsymbol{w} + \boldsymbol{v} || P) + \ln \frac{2n}{\delta} \right)},$$

where $n$ is the size of training set $\mathcal{D}_2$. As we can see, the PAC-Bayes bound in the continual learning scenario depends on four quantities, (1) the stability of old task, (2) the sensitivity of new task, (3) the expected sharpness on training set of new task, and (4) the Kullback Leibler (KL) divergence to the "prior" $P$. The bound is valid for any distribution measure $P$, any perturbation distribution $\boldsymbol{v}$ and any method of choosing $\Delta \boldsymbol{w}$ dependent on the training set $\mathcal{D}_2$.

In order to ensure the stability of old task, we constrain $\Delta \boldsymbol{w}$ in the complementary space $\mathcal{V}^C$ of the important space representing the old task $\mathscr{D}_1$ following GPM [34], so that $\mathbb{E}_{\boldsymbol{v}} [L_{\mathscr{D}_1}(\boldsymbol{w} + \Delta \boldsymbol{w} + \boldsymbol{v})]$ does not increase too much compared with the previously minimized $\mathbb{E}_{\boldsymbol{v}} [L_{\mathscr{D}_1}(\boldsymbol{w} + \boldsymbol{v})]$. Let $\boldsymbol{v} \in \mathcal{V}$, we have $\mathbb{E}_{\boldsymbol{v} \in \mathcal{V}} [L_{\mathcal{D}_2}(\boldsymbol{w} + \Delta \boldsymbol{w} + \boldsymbol{v})] \leq \max_{\boldsymbol{v} \in \mathcal{V}} L_{\mathcal{D}_2}(\boldsymbol{w} + \Delta \boldsymbol{w} + \boldsymbol{v})$, then we can rewrite the above bound as follows:

$$\min_{\Delta \boldsymbol{w}} \mathbb{E}_{\boldsymbol{v}} [L_{\mathscr{D}_1 \cup \mathscr{D}_2}(\boldsymbol{w} + \Delta \boldsymbol{w} + \boldsymbol{v})] \leq \min_{\Delta \boldsymbol{w} \in \mathcal{V}^C} \mathbb{E}_{\boldsymbol{v}} [L_{\mathscr{D}_1}(\boldsymbol{w} + \Delta \boldsymbol{w} + \boldsymbol{v})] + L_{\mathcal{D}_2}(\boldsymbol{w} + \Delta \boldsymbol{w})$$
$$+ \max_{\boldsymbol{v} \in \mathcal{V}} L_{\mathcal{D}_2}(\boldsymbol{w} + \Delta \boldsymbol{w} + \boldsymbol{v}) - L_{\mathcal{D}_2}(\boldsymbol{w} + \Delta \boldsymbol{w}) + 4\sqrt{\frac{1}{n} \left( KL(\boldsymbol{w} + \Delta \boldsymbol{w} + \boldsymbol{v} || P) + \ln \frac{2n}{\delta} \right)}.$$

Thus, our FS-DGPM exactly optimizes the worst-case of the flatness of the weight loss landscape to control the PAC-Bayes bound, which theoretically justifies both lower loss value and flatter landscape lead to better continual learning performance, and why our proposed FS-DGPM works.

## A.2 Derivation for DGPM

We derive the gradient of the importance value $\lambda_i$ by minimizing the worst performance of $f_{\boldsymbol{w}+\boldsymbol{v}}$ under the current task batch $\hat{\mathcal{D}}_t$ and the batch $\hat{\mathcal{M}}$ sampled from the replay buffer $\mathcal{M}$:

$$
\begin{aligned}
\nabla_{\lambda_i} L_{\hat{\mathcal{D}}_t \cup \hat{\mathcal{M}}}(\boldsymbol{w}+\boldsymbol{v}) &= \frac{\partial}{\partial(\boldsymbol{w}+\boldsymbol{v})} L_{\hat{\mathcal{D}}_t \cup \hat{\mathcal{M}}}(\boldsymbol{w}+\boldsymbol{v}) \cdot \frac{\partial}{\partial \lambda_i}(\boldsymbol{w}+\boldsymbol{v}) \\
&= \frac{\partial}{\partial(\boldsymbol{w}+\boldsymbol{v})} L_{\hat{\mathcal{D}}_t \cup \hat{\mathcal{M}}}(\boldsymbol{w}+\boldsymbol{v}) \cdot \frac{\partial}{\partial \lambda_i}\left(\boldsymbol{w}+\eta_1 \sum_{j=1}^{k} \lambda_j \boldsymbol{u}_j \boldsymbol{u}_j^T\left(\nabla_{\boldsymbol{w}} L_{\hat{\mathcal{D}}_t}(\boldsymbol{w})\right)\right) \\
&= \frac{\partial}{\partial(\boldsymbol{w}+\boldsymbol{v})} L_{\hat{\mathcal{D}}_t \cup \hat{\mathcal{M}}}(\boldsymbol{w}+\boldsymbol{v}) \cdot \left(\eta_1 \boldsymbol{u}_i \boldsymbol{u}_i^T\left(\nabla_{\boldsymbol{w}} L_{\hat{\mathcal{D}}_t}(\boldsymbol{w})\right)\right) \\
&\approx \eta_1 \left(\nabla_{\boldsymbol{w}} L_{\hat{\mathcal{D}}_t \cup \hat{\mathcal{M}}}(\boldsymbol{w})\right)^T \boldsymbol{u}_i \boldsymbol{u}_i^T\left(\nabla_{\boldsymbol{w}} L_{\hat{\mathcal{D}}_t}(\boldsymbol{w})\right),
\end{aligned}
$$

where $\cdot$ is the dot product operator. Note that we only consider one gradient update to $\boldsymbol{v}$ in the second equation for simplicity, but using multiple gradient updates is a straightforward extension. For the third equation, we get it by assuming that $\boldsymbol{w}$ is constant with respect to $\lambda_i$. The last approximation is obtained by the first-order Taylor expansion. Setting all first-order gradient terms as constants to ignore second-order derivatives, we get the approximation as:

$$
\nabla_{(\boldsymbol{w}+\boldsymbol{v})} L_{\hat{\mathcal{D}}_t \cup \hat{\mathcal{M}}}(\boldsymbol{w}+\boldsymbol{v}) = \nabla_{\boldsymbol{w}} L_{\hat{\mathcal{D}}_t \cup \hat{\mathcal{M}}}(\boldsymbol{w}) + \left(\nabla_{\boldsymbol{w}}^2 L_{\hat{\mathcal{D}}_t \cup \hat{\mathcal{M}}}(\boldsymbol{w})\right) \boldsymbol{v} + O\left(\|\boldsymbol{v}\|^2\right) \approx \nabla_{\boldsymbol{w}} L_{\hat{\mathcal{D}}_t \cup \hat{\mathcal{M}}}(\boldsymbol{w}).
$$

The importance of each basis is constrained to be between 0 and 1, where 0 indicates that the basis is not important to old tasks and can completely release for learning new tasks. The initial value of all importance is set to 1, and we use the sigmoid function with a temperature factor of 10 at the end of gradient update: $\lambda_i \leftarrow 1/(1 + exp(-10\lambda_i))$.

## A.3 Pseudo-code for updating GPM

GPM [34] achieves excellent stability by ensuring that gradient updates only occur in directions orthogonal to the gradient subspaces deemed important for the past tasks. Similar to [34], we calculate the bases of these subspaces for each layer by analyzing network representations after learning each task with Singular Value Decomposition (SVD), and then use it to update $\boldsymbol{v}$ and $\boldsymbol{w}$ by layer.

As shown in Algorithm 2 for updating GPM, we firstly sample $n_s$ random examples from the replay buffer $\mathcal{M}$ to construct the representation matrix for each layer, $\boldsymbol{R}^l$. Next, we perform SVD on $\boldsymbol{R}^l = \boldsymbol{U}^l \Sigma^l (\boldsymbol{V}^l)^T$ and obtained its $k$-rank approximation $\boldsymbol{R}_k^l$ according to the following **criteria** for the given threshold, $\epsilon^l$:

$$
\|\boldsymbol{R}_k^l\|_F^2 \geq \epsilon^l \|\boldsymbol{R}^l\|_F^2, \tag{7}
$$

where $\|\cdot\|_F$ is the Frobenius norm of the matrix and and $\epsilon^l$ $(0 < \epsilon^l \leq 1)$ is the threshold hyperparameter for layer $l$.

---

**Algorithm 2** UpdateGPM

> **Input:** Network $f_{\boldsymbol{w}}$ with $L$-layer, Replay buffer $\mathcal{M}$, sample size $n_s$, threshold $\epsilon$ for each layer.
> **Result:** Bases matrix $\{(\boldsymbol{M}_l)_{l=1}^{L}\}$           ▷ till $L-1$ if multi-head setting.
> **if** $\mathcal{M}$ is not empty **then**
>      $B_{n_s} \sim \mathcal{M}$           ▷ sample a mini-batch of size $n_s$ from $\mathcal{M}$.
>      $\mathcal{R} \leftarrow$ forward$(B_{n_s}, f_{\boldsymbol{w}})$, where $\mathcal{R} = \{(\boldsymbol{R}^l)_{l=1}^{L}\}$    ▷ construct representation matrix for each
>                                                 layer by forward pass.
>      **for** $l = 1, 2, \cdots, L$ **do**
>          $\boldsymbol{U}^l \leftarrow$ SVD$(\boldsymbol{R}^l)$          ▷ update new bases for each layer by performing SVD.
>          $k \leftarrow$ criteria $(\boldsymbol{R}^l, \epsilon^l)$          ▷ see Eq. (7).
>          $\boldsymbol{M}_l \leftarrow \boldsymbol{U}^l[0:k]$
>      **end for**
> **end if**

---

For fully connected layer, the representation matrix $\boldsymbol{R}^l = [\boldsymbol{x}_1^l, \boldsymbol{x}_2^l, \cdots, \boldsymbol{x}_{n_s}^l]$ concatenates $n_s$ inputs of the $l$-th layer linear function along the column, which are obtained by forwarding the batch of $n_s$ samples $\{\boldsymbol{x}_1, \cdots, \boldsymbol{x}_{n_s}\}$ through the network $f_{\boldsymbol{w}}$. For convolution (Conv) layers, we first express the Conv as matrix multiplication by reshaping the input tensor $\mathcal{X} \in \mathbb{R}^{C_i \times h_i \times w_i}$ and filters $\mathcal{W} \in \mathbb{R}^{C_o \times C_i \times k \times k}$ into $\boldsymbol{X} \in \mathbb{R}^{(h_o \times w_o) \times (C_i \times k \times k)}$ and $\boldsymbol{W} \in \mathbb{R}^{(C_i \times k \times k) \times C_o}$ respectively, where $C_i(C_o)$ denotes the number of input (output) channels of the Conv layer, $h_i, w_i(h_o, w_o)$ represents the height and width of the input (output) feature maps and $k$ is the kernel size of the filters. Then we construct the representation matrix as $\boldsymbol{R}^l = [(\boldsymbol{X}_1^l)^T, (\boldsymbol{X}_2^l)^T, \cdots, (\boldsymbol{X}_{n_s}^l)^T] \in \mathbb{R}^{(C_i \times k \times k) \times (h_o \times w_o \times n_s)}$. The key difference with GPM [34] is that we perform SVD in the entire gradient space and use the obtained bases to replace the last calculated bases, while [34] obtains the newly added bases by performing SVD in the subspace orthogonal to the existing bases. In addition, GPM can be regarded as a special case of our method when $\eta_1$ and $\eta_2$ are set to 0.

# B    Details for Landscape Visualization for Continual Learning

In this section, we first provide the pseudo-code of the visualization of the weight loss landscape in continual learning, and then provide more empirical results.

## B.1    Pseudo-code for Visualization

As shown in Algorithm 3 for the visualization of the weight loss landscape for the continual learning scenario, we first sample a random direction $\boldsymbol{d}$ from a Gaussian distribution, and then apply the filter-wise normalization following [21] to eliminate the scaling invariance of DNNs. Next, we independently calculate the training loss of a series of perturbed weights for each learned task. For a given task descriptor $\tau$ and perturbed weights $\boldsymbol{w} + \alpha \boldsymbol{d}$, we obtain the training loss of the perturbed model $f_{\boldsymbol{w} + \alpha \boldsymbol{d}}$ on all training samples of task $\tau$. Then, we plot the weight loss landscape for task $\tau$. If the descriptor for the current training task of the model is $t$, we will plot $t$ curves.

---

**Algorithm 3** Visualization of the Weight Loss Landscape

---

**Input:** Network $f_{\boldsymbol{w}}$ with $L$-layer ($F_l$ filters in the $l$-th layer), current task descriptor $t$, training dataset $\mathcal{D}_\tau = \{\boldsymbol{x}_{i,\tau}, y_{i,\tau}\}_{i=1}^{n_\tau}$ for $\tau = 1, ..., t$ , the scalar parameter $\alpha \in [\alpha_{min}, \alpha_{max}]$.
Sample a random direction $\boldsymbol{d} \sim \mathcal{N}(0, 1)$
**for** $l = 1, 2, \cdots, L$ **do**
    **for** $j = 1, 2, \cdots, F_l$ **do**
        $\boldsymbol{d}_{l,j} \leftarrow \frac{\boldsymbol{d}_{l,j}}{\|\boldsymbol{d}_{l,j}\|_F} \|\boldsymbol{w}_{l,j}\|_F$            ▷ filter-wise normalization for the random direction.
    **end for**
**end for**
**for** $\tau = 1, 2, \cdots, t$ **do**
    **for** $\alpha = \alpha_{min}, \cdots, \alpha_{max}$ **do**
        $L_{\mathcal{D}_\tau}(\boldsymbol{w} + \alpha \boldsymbol{d}) = \frac{1}{n_\tau} \sum_{(\boldsymbol{x}, y) \in \mathcal{D}_\tau} \ell(f_{\boldsymbol{w} + \alpha \boldsymbol{d}}(\boldsymbol{x}), y)$     ▷ calculate training loss of the
                                                           perturbed model on task $\tau$.
    **end for**
    Plot$(\alpha, L_{\mathcal{D}_\tau}(\boldsymbol{w} + \alpha \boldsymbol{d})), \forall \alpha \in [\alpha_{min}, \alpha_{max}]$
**end for**

---

## B.2    More Results for Section 3.2: Connection of Weight Loss Landscape and Continual Learning

To investigate the relationship between the weight loss landscape and stability-sensitivity in the continual learning scenario, we use the previously proposed GPM [34], La-MAML [13], and ER [5] to train an MLP network with two hidden layers on the Permuted MNIST (PMNIST) [20] dataset that contains 10 tasks. For each task, we use $60,000$ training samples instead of $1,000$ used in the experimental environment. The replay buffer size is set to $1,000$. We also use Oracle and Finetune, which respectively represent retraining the network on the entire dataset contain all passed tasks, and training the network on the data stream without any regularization or episodic memory. Considering the direction $\boldsymbol{d}$ for visualization is randomly selected, we repeat the visualization 10 times with different $\boldsymbol{d}$. Figure 10, 11, 12, 13, and 14 show the weight loss landscape for each task when a new

task is trained using Oracle, Finetune, GPM, La-MAML, and ER, respectively. The $i$-th row of each figure represents changes in the weight loss landscape of the $i$-th task during the model evolution, and each $j$-th column indicates that the current model has learned $j$ tasks.

## C    Details for FS-ER

In this section, we first provide the pseudo-code of Flattening Sharpness for Vanilla ER, and then provide more empirical results.

### C.1    Pseudo-code for FS-ER

Comparing the pseudo-code of Vanilla ER, FS-ER only adds the adversarial weight perturbation $v$.

---
**Algorithm 4** FS-ER

---
    **Input:** Network weight $w$, loss function $\ell$, learning rate $\eta_3$, FS step size $\eta_1$, FS steps $K$, batch size $b$.
    Initializing $\mathcal{M} \leftarrow \{\}$
    **for** $t = 1, 2, \cdots, T$ **do**
        **for** $ep = 1, 2, \cdots, num_{epochs}$ **do**
            **for** batch $\hat{\mathcal{D}}_t \overset{b}{\sim} \mathcal{D}_t$ **do**        $\triangleright$ Sample without replacement a mini-batch from task $t$.
                $\hat{\mathcal{M}} \overset{b}{\sim} \mathcal{M}$                $\triangleright$ Sample a mini-batch from $\mathcal{M}$.
                **for** $k = 1, \cdots, K$ **do**
                    $v \leftarrow v + \eta_1 \nabla_{(w+v)} L_{\hat{\mathcal{D}}_t \cup \hat{\mathcal{M}}}(w + v)$    $\triangleright$ Add adversarial weight perturbation $v$.
                **end for**
                $w \leftarrow w - \eta_3 \nabla_w L_{\hat{\mathcal{D}}_t \cup \hat{\mathcal{M}}}(w + v)$
                Push $\hat{\mathcal{D}}_t$ to $\mathcal{M}$ with reservior sampling
            **end for**
        **end for**
    **end for**

---

### C.2    More Results for Section 3.3: A Case Study of Flattening Sharpness for Vanilla ER

Figure 14 and 15 show the weight loss landscape for each task when a new task is trained using ER and FS-ER, respectively.

## D    Experimental Details

### D.1    Datasets

Table 4 summarizes the statistics of four datasets used in our experiments.

Table 4: Dataset Statistics.

|  | **PMNIST** | **CIFAR-100 Split** | **CIFAR-100 Superclass** | **TinyImageNet** |
|---|---|---|---|---|
| Input size | $1\times28\times28$ | $3\times32\times32$ | $3\times32\times32$ | $3\times64\times64$ |
| # tasks | 20 | 10 | 20 | 40 |
| # Classes/task | 10 | 10 | 5 | 5 |
| # Training/task | 1,000 | 4,750 | 2,375 | 2,250 |
| # Validation/task | - | 250 | 125 | 250 |
| # Test/task | 10,000 | 1,000 | 500 | 250 |

## D.2 Architecture

**AlexNet-like architecture:** For 10-Split CIFAR-100, similar to GPM [34], we use a AlexNet-like architecture with three convolutional (Conv) layers and two fully connected layers. The three Conv layers have 64, 128, and 256 filters with 4×4, 3×3, and 2×2 kernel sizes, respectively. Both fully connected layers have 2048 units in each layer. Max-pooling layer with filters of size 2×2 is used after the Conv layer. Dropout of 0.2 is used for the first two layers, and 0.5 is used for the remaining layers. Batch normalization is only used in the second layer.

**Modified Lenet-5 architecture:** For 20-Spilt CIFAR-100 Superclass, similar to GPM [34], we use a modified LeNet-5 architecture with 20-50-800-500 neurons, of which the first two are Conv layers with 5×5 kernel sizes, and the last two are fully connected layers. Batch normalization and max-pooling layer with filters of size 3×3 with a stride of 2 are used in the Conv layers. Batch normalization parameters are learned for the first task and shared with all the other tasks.

**Architecture for TinyImageNet:** For 40-Spilt TinyImageNet, similar to La-MAML [13], we use a CNN having 4 Conv layers with 160 3×3 filters. The output from the final Conv layer is flattened and is passed through 2 fully connected layers having 320 and 640 units, respectively.

All networks use ReLU in the hidden units, and finally have a multi-head output layer to perform classification for every task. No bias units are used following [34].

## D.3 Baselines

We compare our method against multiple methods described below.

- **EWC** [17]: Elastic Weight Consolidation is a regularization-based method that uses the diagonal of Fisher information to identify important weights.

- **ICARL** [30]: ICARL is a memory-based method that uses knowledge distillation and episodic memory to reduce forgetting.

- **GEM** [24]: Gradient Episodic Memory uses the gradient of episodic memory to constrain optimization to make sure that the gradients of the new task do not change the previous knowledge.

- **ER** [31, 5]: Experience Replay uses a small replay buffer to store old data using reservoir sampling. Then, the stored data is replayed again with the new data samples.

- **La-MAML** [13]: Look-ahead MAML is inspired by optimization-based meta-learning that leverages replay to avoid forgetting and favor positive backward transfer by asynchronously learning the weights and LRs.

- **GPM** [34]: Gradient Projection Memory minimizes forgetting by taking gradient steps orthogonal to the gradient subspace deemed important for the past tasks when learning a new task.

- **Multitask**: Multitask is an oracle baseline that all tasks are learned jointly using the entire dataset at once in a single network. Multitask is not a continual learning strategy but serves as an upper bound on average test accuracy on all tasks.

GEM [24], ICARL [30], and La-MAML [13] are implemented from the official implementation provided by [13]. EWC [17] is implemented from the official implementation provided by [24]. GPM [34] is implemented from its official implementation. ER [31, 5] is implemented by adapting the code provided by [31].

## D.4 GPU Device

We measured the average training time per task calculated on the NVIDIA GeForce RTX 2080 Ti GPU. Figure 5(b) shows the training time per task on the 20-Split CIFAR-100 Superclass experiment using the baselines and our method.

### D.5 Threshold Hyperparameter

Following [34], we use a different value of the threshold hyperparameter, $\epsilon$, for different architectures and different datasets. For PMNIST experiment, we use $\epsilon = 0.99$ for all the layers and increasing the value of $\epsilon$ by 0.0005 for each new task. For CIFAR-100 Split and CIFAR-100 Superclass experiment, we use the values reported by [34]. For CIFAR-100 Split, the initial value of $\epsilon$ is 0.97 for all the layers and increasing by 0.003 for each new task. For CIFAR-100 Superclass experiment, we use $\epsilon = 0.98$ for all the layers and increasing by 0.001 for each new task. For TinyImageNet experiment, we use $\epsilon = 0.9$ for all the layers and increasing by 0.0025 for each new task.

### D.6 List of Hyperparameters

We report in Table 5 the hyper-parameters selected by grid-search for all baselines and our method. For PMNIST, we use a hyper-parameter called glances for all compared approaches and set it to 5 following [13]. This hyper-parameters indicates the number of meta-updates made on each incoming sample of data. In addition, for each task in PMNIST, we use 5 epochs to train the network for Multitask instead of 1 epoch in other methods.

Table 5: List of hyperparameters for the baselines and our approach. "LR" denotes the (initial) learning rate. Superclass is the abbr. of CIFAR-100 Superclass.

| Method | Parameter | PMNIST | CIFAR-100 Split | Superclass | TinyImageNet |
|---|---|---|---|---|---|
| EWC | LR | 0.01 | 0.005 | 0.03 | - |
| | memory strength, $\gamma$ | 100 | 5000 | 1000 | - |
| ICARL | LR | - | 0.03 | 0.01 | - |
| | memory strength, $\gamma$ | - | 0.1 | 0.5 | - |
| | memory size | - | 1000 | 1000 | - |
| GEM | LR | 0.01 | 0.01 | 0.03 | - |
| | memory strength, $\gamma$ | 0.0 | 0.5 | 0.5 | - |
| | memory size | 200 | 1000 | 1000 | - |
| ER | LR | 0.005 | 0.03 | 0.01 | - |
| | memory size | 200 | 1000 | 1000 | - |
| La-MAML | LRs, $\alpha_0$ | 0.15 | 0.1 | 0.1 | - |
| | LR for LRs, $\eta$ | 0.3 | 0.5 | 0.5 | - |
| | memory size | 200 | 1000 | 1000 | - |
| GPM | LR | 0.01 | 0.01 | 0.01 | 0.005 |
| | $n_s$ | 300 | 125 | 125 | 200 |
| Multitask | LR | 0.01 | 0.01 | - | - |
| FS-DGPM | LR, $\eta_3$ | 0.01 | 0.01 | 0.01 | 0.01 |
| | LR for sharpness, $\eta_1$ | 0.05 | 0.001 | 0.01 | 0.001 |
| | LR for DGPM, $\eta_2$ | 0.01 | 0.01 | 0.01 | 0.01 |
| | memory size | 200 | 1000 | 1000 | 400 |
| | $n_s$ | 200 | 125 | 125 | 200 |

## E Additional Experimental Results

We provide the results of different step numbers $K$ in solving weight perturbation $\boldsymbol{v}$. We evaluation FS-DGPM with $K \in \{1, 2, 3\}$ in the CIFAR-100 Split experiment. As shown in Figure 7, two steps have been well improved, and the extra steps only bring few improvements but with much more time.

We also compare ER and our method on CIFAR-100 Superclass when the memory size is 100, 500, and 1000. As shown in Figure 8, we see that both ER and FS-DGPM benefits from increases in memory size, but the outperformance of FS-DGPM is more visible under the low-resource regime. We think the advantageous performance of FS-DGPM can be attributed to the effective utilization of episodic memory converted into bases through SVD.

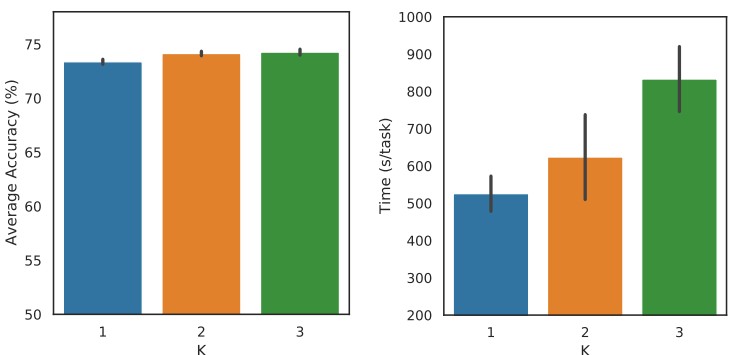

Figure 7: Results of different step numbers $K$ for weight perturbation $v$ on CIFAR-100 Split in 50 epochs. Each experiment is run with 5 seeds.

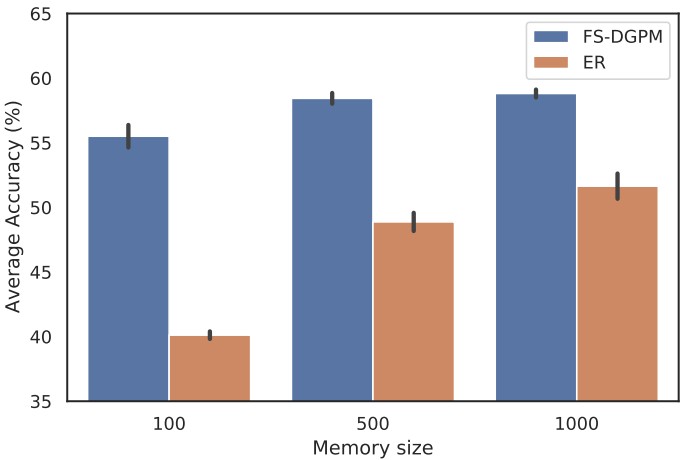

Figure 8: Average test accuracy of different memory size on CIFAR-100 SuperClass. Each experiment is run with 5 seeds.

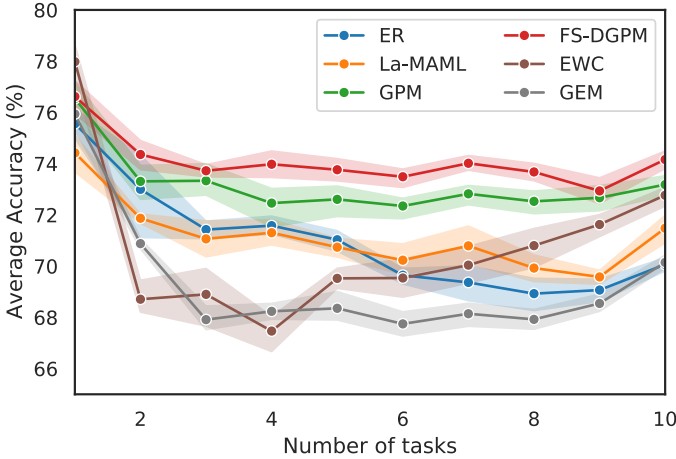

Figure 9: Average test accuracy as a function of the number of tasks trained on 10-Split CIFAR-100. Each experiment is run with 5 seeds.

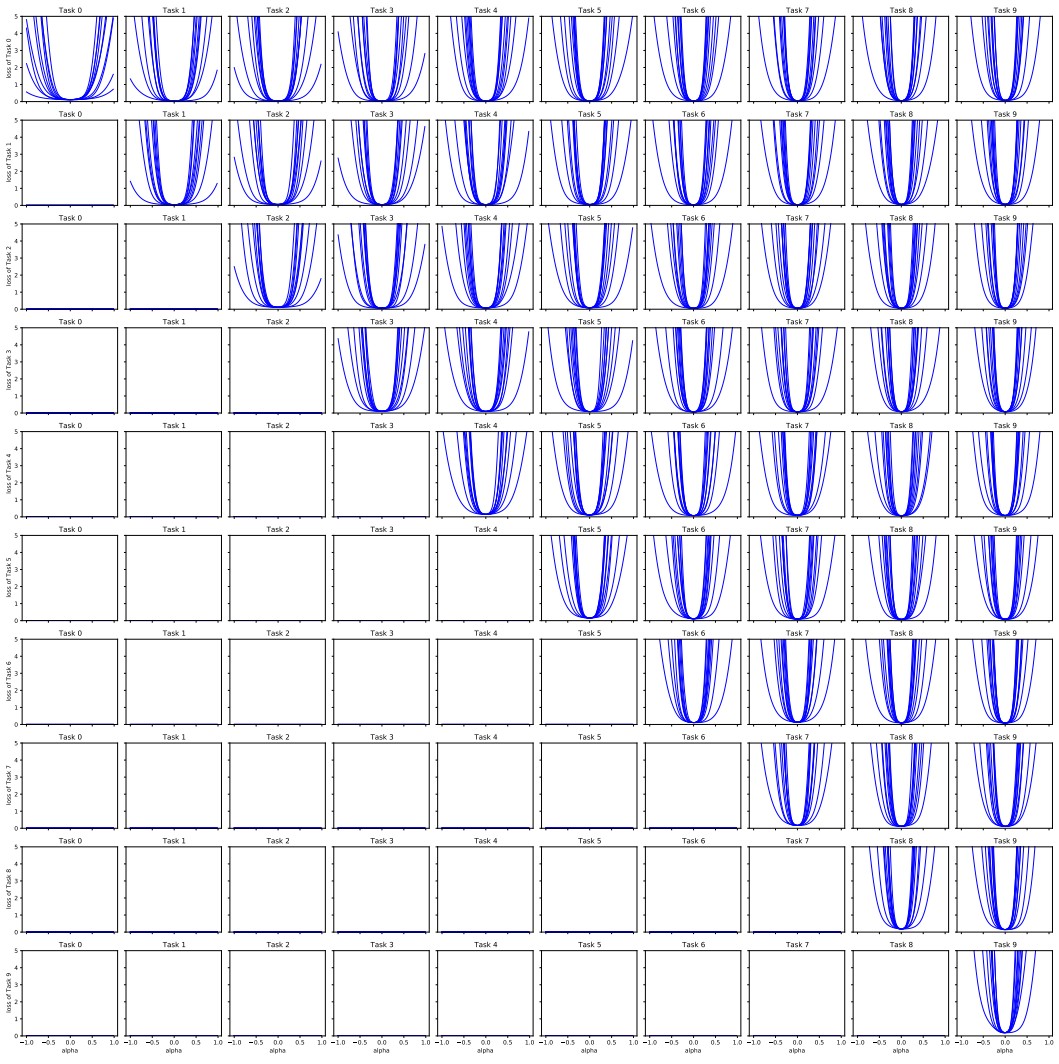

Figure 10: The weight loss landscape of the ten tasks of PMNIST using Oracle for training. The $i$-th row represents changes in the weight loss landscape of the $i$-th task during the model evolution, and the $j$-th column indicates that the model has learned $j$ tasks. The y-axis is the loss value, and the x-axis is the scalar value for visualization random direction. (Task $t$ is the abbr. of $t + 1$-th task.)

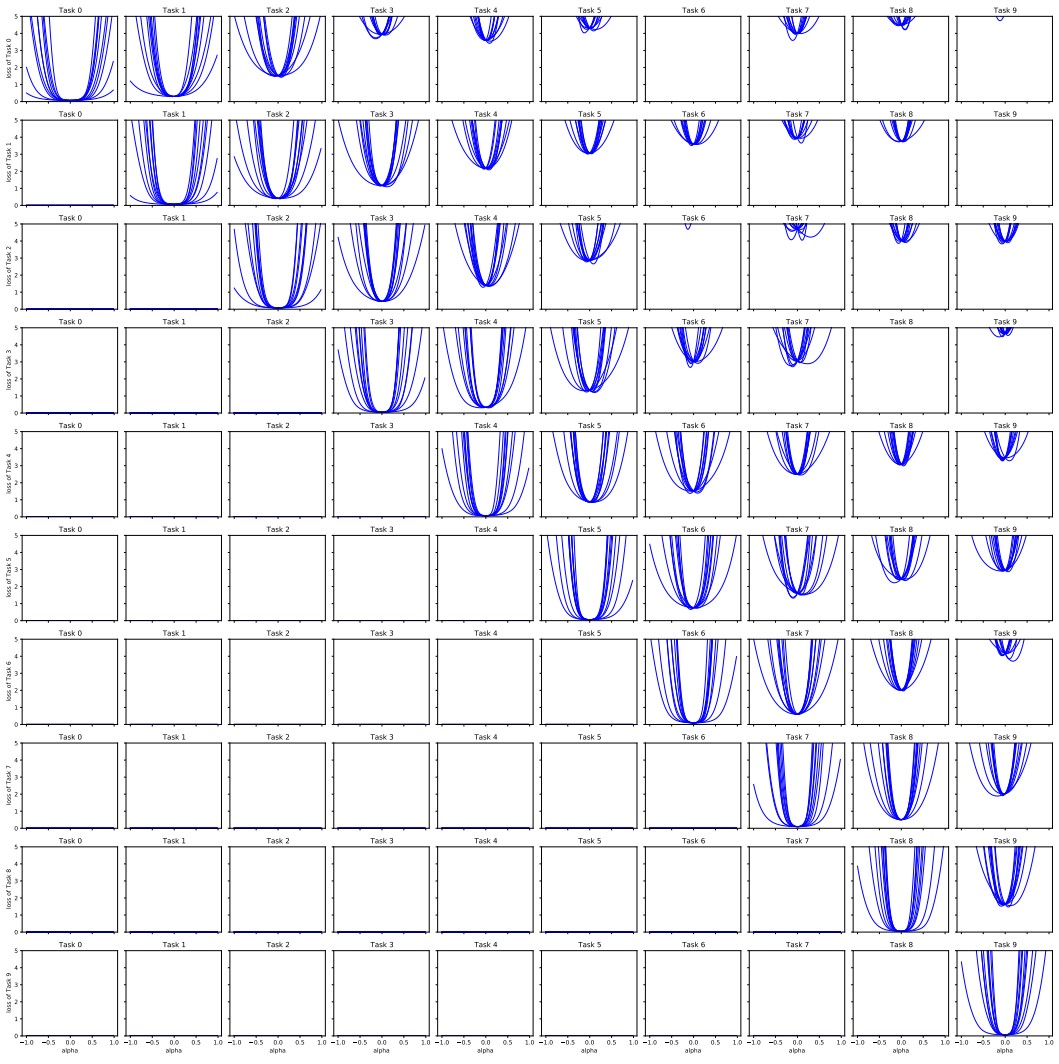

Figure 11: The weight loss landscape of the ten tasks of PMNIST using Finetune for training. The $i$-th row represents changes in the weight loss landscape of the $i$-th task during the model evolution, and the $j$-th column indicates that the model has learned $j$ tasks. The y-axis is the loss value, and the x-axis is the scalar value for visualization random direction. (Task $t$ is the abbr. of $t + 1$-th task.)

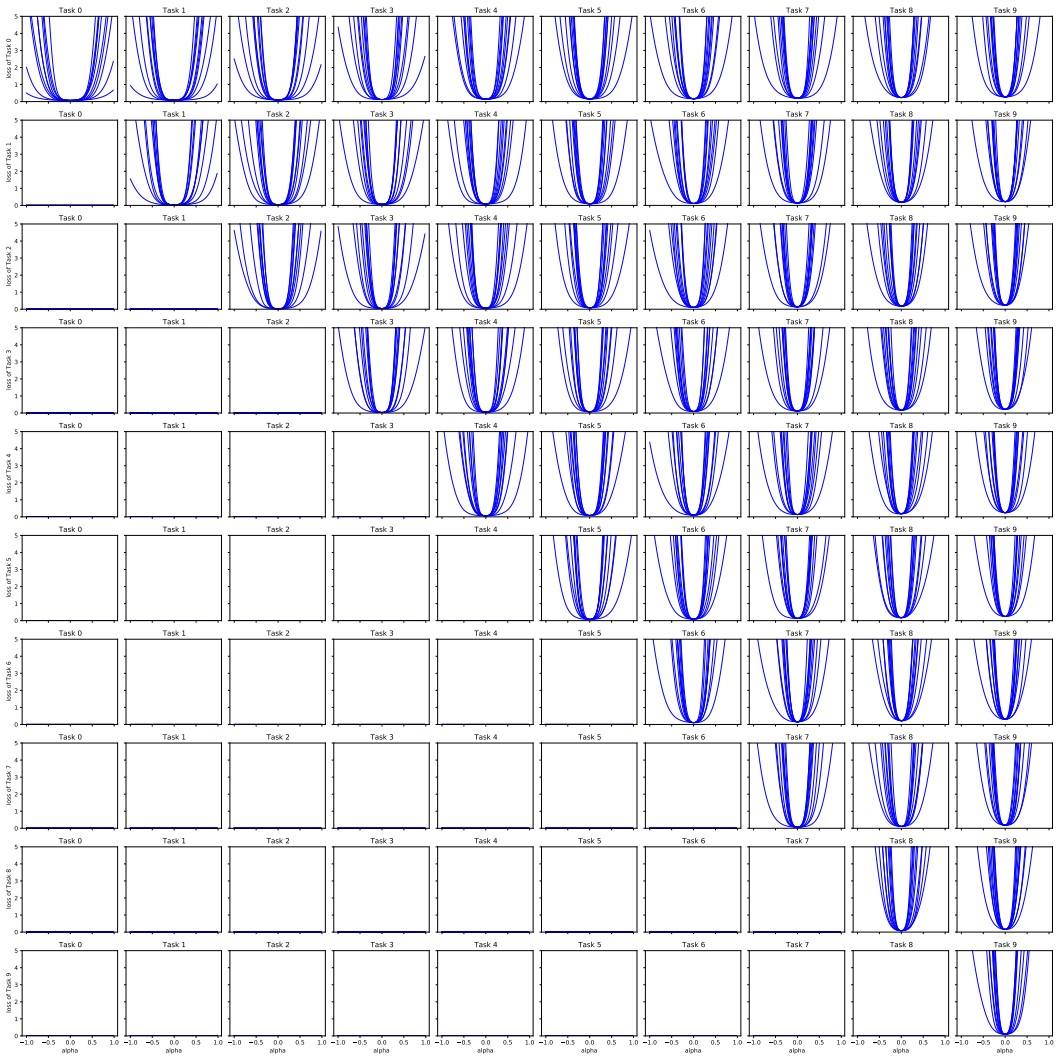

Figure 12: The weight loss landscape of the ten tasks of PMNIST using GPM for training. The $i$-th row represents changes in the weight loss landscape of the $i$-th task during the model evolution, and the $j$-th column indicates that the model has learned $j$ tasks. The y-axis is the loss value, and the x-axis is the scalar value for visualization random direction. (Task $t$ is the abbr. of $t + 1$-th task.)

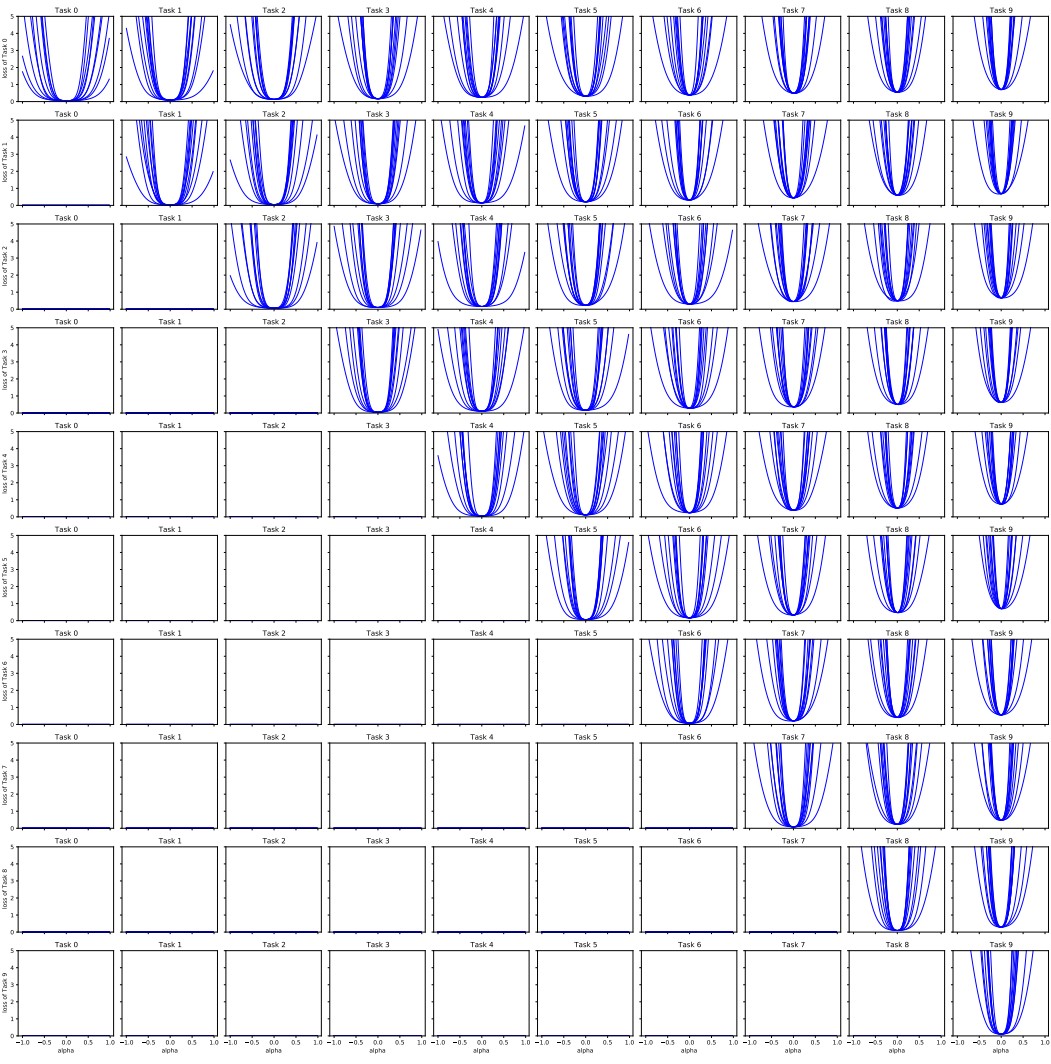

Figure 13: The weight loss landscape of the ten tasks of PMNIST using La-MAML for training. The $i$-th row represents changes in the weight loss landscape of the $i$-th task during the model evolution, and the $j$-th column indicates that the model has learned $j$ tasks. The y-axis is the loss value, and the x-axis is the scalar value for visualization random direction. (Task $t$ is the abbr. of $t+1$-th task.)

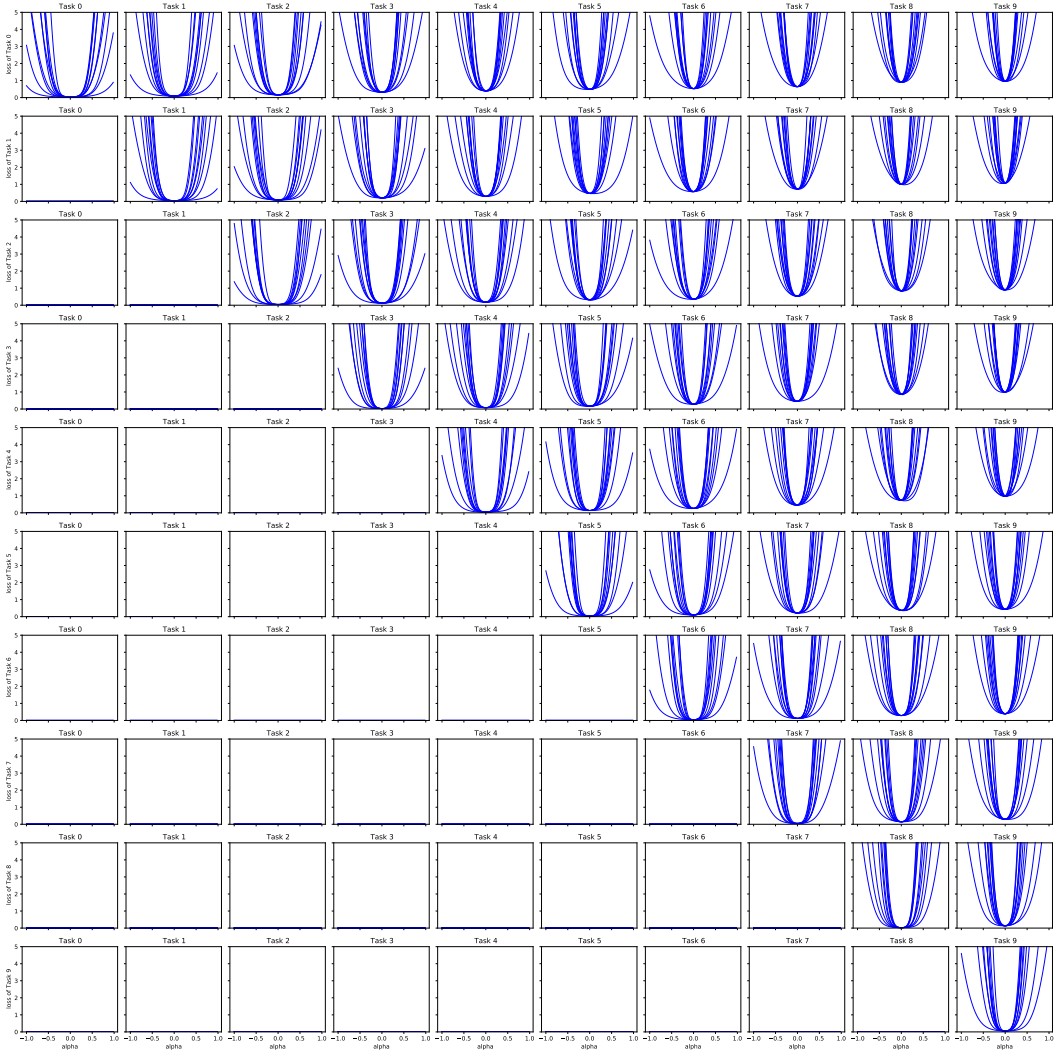

Figure 14: The weight loss landscape of the ten tasks of PMNIST using ER for training. The $i$-th row represents changes in the weight loss landscape of the $i$-th task during the model evolution, and the $j$-th column indicates that the model has learned $j$ tasks. The y-axis is the loss value, and the x-axis is the scalar value for visualization random direction. (Task $t$ is the abbr. of $t+1$-th task.)

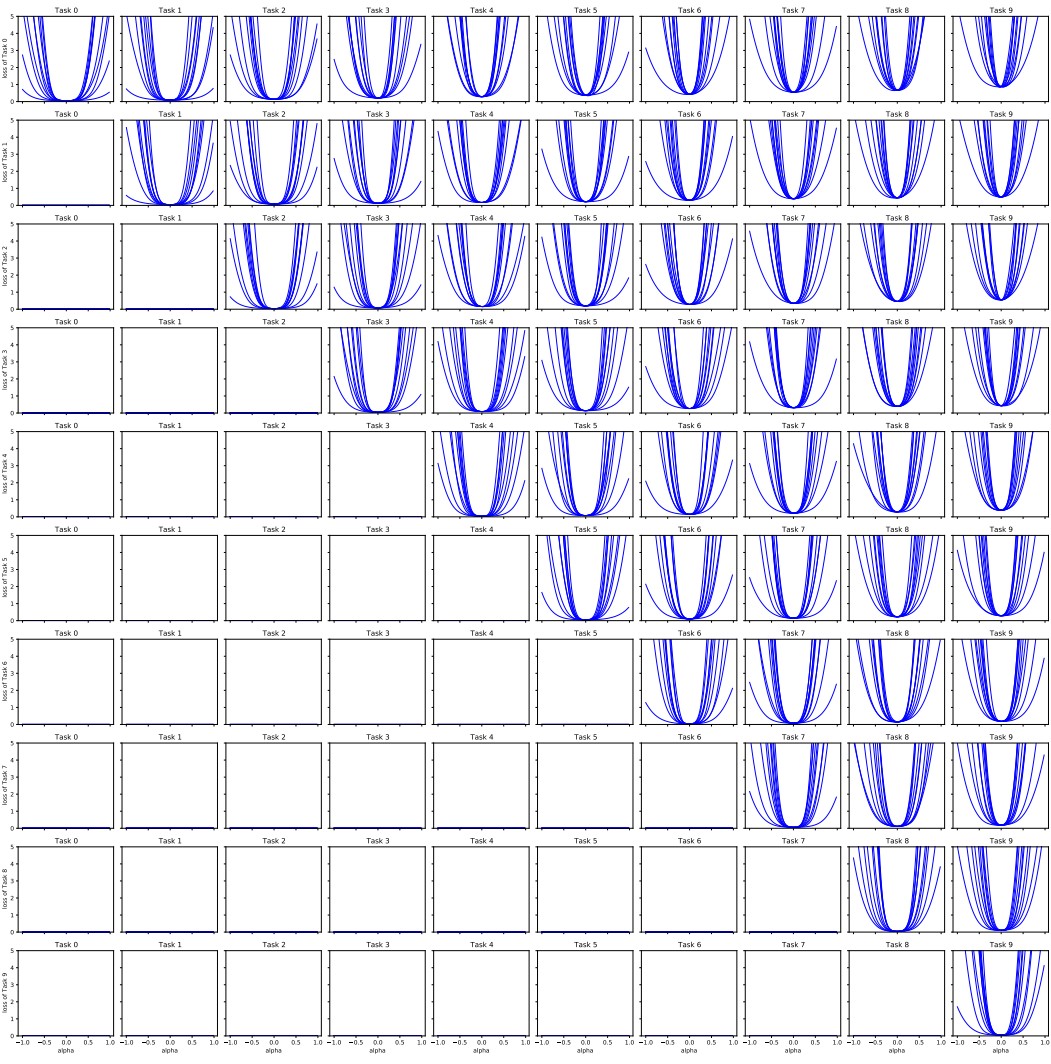

Figure 15: The weight loss landscape of the ten tasks of PMNIST using FS-ER for training. The $i$-th row represents changes in the weight loss landscape of the $i$-th task during the model evolution, and the $j$-th column indicates that the model has learned $j$ tasks. The y-axis is the loss value, and the x-axis is the scalar value for visualization random direction. (Task $t$ is the abbr. of $t+1$-th task.)