# OpenReview forum: "Flattening Sharpness for Dynamic Gradient Projection Memory Benefits Continual Learning"
_NeurIPS.cc/2021/Conference — NeurIPS 2021 Poster_

### Official Review · Reviewer_hs57 · 2021-07-14

**Rating:** 6
**Confidence:** 4

**Summary:**

In this paper, the authors propose a method called Flattening Sharpness for Dynamic Gradient Projection Memory (FS-DGPM) for continual learning. The main idea is to investigate the effect of weight loss landscape on overcoming catastrophic forgetting. With the understanding of the weight loss landscape, the authors further propose a method based on the recently proposed Gradient Projection Memory (GPM) method to predict the importance of bases which span the subspace of old tasks. Less important bases can be dynamically released based on the proposed soft weight. Extensive experiments show that the proposed method can consistently outperform the state-of-the-art methods across a range of widely used benchmark datasets.

**Limitations And Societal Impact:**

Yes

**Main Review:**


Originality: The idea of leveraging the flatness of the minima for continual learning is novel and interesting. However, the proposed method seems incremental.

Quality: The paper is technically sound.


Clarity: The writing of the paper needs improvement. The paper should include a clear introduction to GPM such that it is easier to understand the proposed method. Meanwhile, the proposed method is not clearly explained. There are also several typos in the paper.


Significance: The improvements are not significant but meaningful which show the effectiveness of utilizing the loss landscape.



Questions and suggestions:


1. Can the idea of leveraging the loss landscape be used for other continual learning methods other than GPM?

2. To show the effect of loss landscape on continual learning, maybe the authors should conduct a simpler experiment to remove the factor of continual learning methods themselves. For example, just train a vanilla network sequentially on all the tasks, then compare the performance of the network with or without enforcing flat minima.


**Time Spent Reviewing:**

3 hours

---

> ### Author Response · Authors · 2021-08-06
> **Author's Response to Reviewer hs57**
>
> We thank the reviewer for the review and comments on our paper. We will polish our manuscript by adding a clear introduction of GPM. We address the reviewer’s comment below:
>
> **Reviewer’s Comment 1:** “Can the idea of leveraging the loss landscape be used for other continual learning methods other than GPM?”
> **Response:** The idea of flattening sharpness is model-agnostic, which can be applied to other continual learning methods only by adding an adversarial weight perturbation before updating weight. As shown in Section 3.3, the development of our FS-DGPM starts with performing a case study of flattening sharpness for Vanilla ER, which obtains a significant improvement in the average testing accuracy from **86.16%** to **90.44%**.
>
> **Reviewer’s Comment 2:** “To show the effect of loss landscape on continual learning, maybe the authors should conduct a simpler experiment to remove the factor of continual learning methods themselves. For example, just train a vanilla network sequentially on all the tasks, then compare the performance of the network with or without enforcing flat minima.”
> **Response:** Thanks for your suggestion. We have actually performed a fine-tuning experiment (just training a vanilla network sequentially on all the tasks, as requested). However, due to the serious forgetting of the old task, the loss value of Task1 after learning Task10 using fine-tuning is much greater than that of using CL algorithms (**Fine-tuning : GPM : ER : La-MAML= 5.3 : 0.3 : 0.9 : 0.7**). In order to better visualize and compare the performance of CL methods, we didn't display the effect of fine-tuning in Figure 1 of the main paper. We will add the weight loss landscapes using fine-tuning in our revised appendix.

---

### Official Review · Reviewer_sRK3 · 2021-07-15

**Rating:** 7
**Confidence:** 4

**Summary:**

The paper studies catastrophic forgetting and continual learning from the loss landscape perspective.
The primary motivation is that from the plasticity/stability view, stable networks tend to have flatter minima.
While this has been observed before in [1], the paper studies the problem in more depth. Also, it proposes a solution to reduce the sharpness of minima for each task based on the gradient projection memory work.
Overall, I like the paper, and I think it is an influential continual learning paper. However, I think the paper can be even better if the authors can address a few shortcomings.

**Limitations And Societal Impact:**

Yes

**Main Review:**


### Overall
**Originality**: The significance of minima sharpness has been studied before in CL literature [1], but the work is novel enough regarding the depth and the approach of analysis.
**Significance**: The paper is a significant contribution since it gives a better understanding of catastrophic forgetting and continual learning in general.
**Quality**: I believe this is a high-quality work both theoretically and empirically.
**Clarity**: The paper is well-written, aside from a few problems regarding the appendix (see below).

----

### Strengths
**1-** I appreciate the fact that this paper aims to increase our understanding of catastrophic forgetting. In my opinion, the CL literature needs more works that try to understand the continual learning challenges.
**2-** The proposed method and analysis seem novel and reasonable.
**3-** The experiments are very well designed. However, they can be improved to make the contributions of the paper even more significant.


### Weaknesses
**1-** Since the main objective of the proposed method is to reduce the sharpness of minima, and since the authors seem to be familiar with the impact of training regime on sharpness [1], I wonder why they have chosen to use a single hyper-parameter rather than integrate the findings of [1] regarding the training regime. Can the authors elaborate on this? Aside from that, I think it will be fairer if the results of Tables 1 and 2 were calculated using a grid-search over different hyper-parameters since using only one set of parameters might favor a specific method.

**2-** Figure 6 illustrates the effectiveness of FS-DGPM over other methods and shows that around the local minimizer, the loss landscape of FS-DGPM seems to be flatter than other methods. But, what about the Multitask method? What would the results look like compared to the upper-bound, which is the multitask training? Could the authors report the results in this regard?

**3-** While Figure 5 (c) shows the memory usage, it is unclear what would happen if the memory budget increases/decreases. Could the authors perform an experiment where they compare the results for ER and FS-DGPM when the memory size increases? For instance, an experiment similar to figure 1 in [2] can show the behavior of FS-DGPM over different scenarios. For instance, for Split CIFAR-100 (super), memory sizes of 100, 500, 1000 seem representative enough.

**4-** It seems like the appendix is not well-connected to some parts of the main text. For instance, in Section D.4 (L135), the paper mentions figure 5 (b) while it refers to figure 5 of the main text rather than figure 5 of the paper. Also, in the main text (L283), the paper mentions the training details hyperparameters are reported in Appendix D.4 while they are discussed in Appendix D.5 and D.6. Also, Figure 1 in the appendix does not have a legend.

**5-** Do authors plan to release their code for this paper?

----- Update after discussion period ------
The authors have addressed my comments and I still vote accepting the paper.


[1] *Mirzadeh, Seyed Iman, et al. "Understanding the Role of Training Regimes in Continual Learning." Advances in Neural Information Processing Systems, 2020*
[2] *Chaudhry, Arslan, et al. Continual Learning with Tiny Episodic Memories. 2019. https://arxiv.org/pdf/1902.10486.pdf*

**Time Spent Reviewing:**

4

---

> ### Author Response · Authors · 2021-08-06
> **Author's Response to Reviewer sRK3 (2/2)**
>
> **Reviewer’s Comment 3:** “While Figure 5 (c) shows the memory usage, it is unclear what would happen if the memory budget increases/decreases. Could the authors perform an experiment where they compare the results for ER and FS-DGPM when the memory size increases? ... For instance, for Split CIFAR-100 (super), memory sizes of 100, 500, 1000 seem representative enough.”
> **Response:** We thank the reviewer for suggesting the memory budget experiment. According to the reviewer’s suggestion, we added new experiments to compare ER and FS-DGPM on Split CIFAR-100 (super) when the memory size is 100, 500, and 1000. The results are given below, and each experiment is run with 5 seeds:
>
> + **Average Accuracy (ACC)**
>
> | Methods\Mem_size | 1000 | 500 | 100 |
> | :----  | :----:  | :----:  | :----:  |
> | ER | 51.64% $\pm$ 1.09 | 48.88% $\pm$ 0.79 | 40.11% $\pm$ 0.33 |
> | FS-DGPM | **58.81% $\pm$ 0.34** | **58.44% $\pm$ 0.47** | **55.51% $\pm$ 0.98** |
> | GAP=(FS-DGPM - ER)/ER | 13.88% | 19.56% | 38.38% |
>
> + **Backward Transfer (BWT)**
>
> | Methods\Mem_size | 1000 | 500 | 100 |
> | :----  | :----:  | :----:  | :----:  |
> | ER | -7.86% $\pm$ 0.89 | -10.83% $\pm$ 0.71 | -19.56% $\pm$ 1.01 |
> | FS-DGPM | **-2.97% $\pm$ 0.35** | **-3.15% $\pm$ 0.40** | **-6.28% $\pm$ 1.00** |
>
> As shown in the above results, we see that both ER and FS-DGPM benefits from increases in memory size, **but the outperformance of FS-DGPM is more visible under the low-resource regime**. We think the advantageous performance of FS-DGPM can be attributed to the effective utilization of episodic memory converted into bases through SVD.
>
> **Reviewer’s Comment 4:** “It seems like the appendix is not well-connected to some parts of the main text. For instance, in Section D.4 (L135), the paper mentions figure 5 (b) while it refers to figure 5 of the main text rather than figure 5 of the paper. Also, in the main text (L283), the paper mentions the training details hyperparameters are reported in Appendix D.4 while they are discussed in Appendix D.5 and D.6. Also, Figure 1 in the appendix does not have a legend.”
> **Response:** We thank the reviewer for pointing out these typos, and will polish our manuscript. For the legend of Figure 1 in the appendix, different color indicates the different step number K, which has been displayed on the horizontal axis, so the legend is not included in the figure.
>
>
> **Reviewer’s Comment 5:** “Do authors plan to release their code for this paper?”
> **Response:** Yes, we will include the GitHub link of our code in the camera-ready version.

---

> ### Author Response · Authors · 2021-08-06
> **Author's Response to Reviewer sRK3 (1/2)**
>
> We thank the reviewer for taking the time and reviewing our paper. We are glad to hear his/her positive comments and appreciation for our work. We provide our point-by-point responses below:
>
> **Reviewer’s Comment 1**: “Since the main objective of the proposed method is to reduce the sharpness of minima, and since the authors seem to be familiar with the impact of training regime on sharpness [1], I wonder why they have chosen to use a single hyper-parameter rather than integrate the findings of [1] regarding the training regime. Can the authors elaborate on this? Aside from that, I think it will be fairer if the results of Tables 1 and 2 were calculated using a grid-search over different hyper-parameters since using only one set of parameters might favor a specific method.”
> **Response:** We thank the reviewer for this suggestion. In our experiments, we consider the CIFAR-100 (Split and Super) and PMNIST/TinyImageNet datasets, and follow the hyper-parameters reported in GPM [2] and La-MAML [3] for fair comparisons, such as the network architecture, batch size, optimizer, etc. We have conducted experiments using a grid-search over the learning rate in our original submission. As per the reviewer’s suggestion, we added new experiments using a grid-search over batch size and learning rate jointly on CIFAR-100 (Split). Following [1], we consider two batch sizes with 10 and 64. The training time of the batch size of 10 is more than five times that of 64, so we run each experiment with 3 seeds due to the computational resource limitation. Results are given below:
>
> + **Experiment 1: Batch_size=10**
>
> | Methods\learning_rate |  0.03 |  0.01 |  0.005 |
> |  :----  | :----:  | :----:  | :----:  |
> | ER | 62.94% $\pm$ 1.09 | 69.85% $\pm$ 0.50 | **70.99% $\pm$ 0.17** |
> | GPM |64.90% $\pm$ 0.46 | 71.93% $\pm$ 0.60 | **72.11% $\pm$ 0.53** |
> | FS-DGPM | 67.74% $\pm$ 0.24  | 72.53% $\pm$ 0.59 | **74.69% $\pm$ 0.65** |
>
>
> + **Experiment 2: Batch_size=64**
>
> | Methods\learning_rate |  0.03 |  0.01 |  0.005 |
> |  :----  | :----:  | :----:  | :----:  |
> | ER | 70.07% $\pm$ 0.35 | **71.53% $\pm$ 0.30** | 68.67% $\pm$ 0.37 |
> | GPM | 71.77% $\pm$ 0.35 | **73.18% $\pm$ 0.52** | 70.42% $\pm$ 0.29 |
> | FS-DGPM | 72.07% $\pm$ 0.18 | **74.33% $\pm$ 0.31** | 70.91% $\pm$ 0.64 |
>
> As shown in the above results, we see that different hyper-parameter lead to a different performance in accordance with [1], and our method outperforms the other two methods under the different choices of batch size.
>
>
> **Reviewer’s Comment 2:** “Figure 6 illustrates the effectiveness of FS-DGPM over other methods and shows that around the local minimizer, the loss landscape of FS-DGPM seems to be flatter than other methods. But, what about the Multitask method? What would the results look like compared to the upper-bound, which is the multitask training? Could the authors report the results in this regard?”
> **Response:** Thanks for the reviewer's suggestion. We will update Figure 6 in our revised manuscript by adding the loss landscape of Multitask method. We briefly summarize the empirical results below:
>
> **Experiment 1: The stability of the first task**
>
> (a) **Test accuracy change of the first task**
>
> | Methods\Number of tasks | 1 | 5 | 10 |
> |  :----  | :----:  | :----:  | :----:  |
> | FS-DGPM | 76.86% | 77.8% | 78.5% |
> | Multitask | 75.6% | 82.6% | 82.6% |
>
> (b) **The weight loss landscape of Task1 after learning Task10**
> The range of loss values for Task1 within the unit neighborhood after learning Task10 are listed as follows:
>
> | Methods | Range of loss values ($[loss_{min}, loss_{max}]$)|
> |  :----  | :----: |
> | FS-DGPM | [0.3, 298] |
> | Multitask | [0.07, 294] |
>
> The Multitask method shows a better stability capability with **a lower loss value and a flatter loss landscape** than FS-DGPM.
>
> **Experiment 2: The sensitivity of the fifth task**
>
> (a) **Test accuracy of the first task after learning Task 5**
>
> | Methods | ACC |
> |  ----  | :---- |
> | FS-DGPM | 77.3% |
> | Multitask | 79.4% |
>
> (b) **The weight loss landscape of Task5 after learning Task 5**
>
> | Methods | Range of loss values ($[loss_{min}, loss_{max}]$)|
> |  :----  | :----: |
> | FS-DGPM | [0.14, 210] |
> | Multitask | [0.15, 96] |
>
> We can see that using the Multitask method has **a flatter loss landscape** than FS-DGPM, even though its minimal loss value is slightly higher than FS-DGPM.
>
> [1] *Mirzadeh S I, Farajtabar M, Pascanu R, et al. Understanding the Role of Training Regimes in Continual Learning. NeurIPs, 2020*
> [2] *Saha G, Garg I, Roy K. Gradient projection memory for continual learning, ICLR 2021.*
> [3] *Gupta G, Yadav K, Paull L. La-maml: Look-ahead meta learning for continual learning. NeurIPs, 2020.*

---

> > ### Comment · Reviewer_sRK3 · 2021-09-03
> > **Response to authors**
> >
> > Thank you for your detailed response. I believe my comments have been addressed.

---

### Official Review · Reviewer_tT1e · 2021-07-15

**Rating:** 6
**Confidence:** 5

**Summary:**

The paper proposes a new continual learning method that promotes the loss shapes to be flattened. The main idea is to optimize the worst case perturbed loss function at every gradient step, and the empirical investigation of the loss values around the learned parameter corroborates their assertion. Also, experimentally, they showed promising results compared to other recent baselines.

**Main Review:**

First of all, I think the paper misses an important reference that also had a similar motivation of attaining the flat minima, or the wide local minima (WLL), in the context of continual learning:

Cha. et al., CPR: Classifier-projection regularization for continual learning, ICLR 2021.

While the approach is different, I think it definitely need to be cited.

Having said that, the proposed method of the paper seems novel in the context of attaining flat minima in the continual learning context. Followings are what I think about the paper:

Pros:
(+) Framework of minimax optimization and empirically showing that the loss function becomes flat seems to be novel.
(+) Provided theoretical explanations, although the connection with the proposed method is not very straightforward.
(+) Experimental results tend to be thorough, providing ablation study and memory usage, time complexity comparisons.

Cons:
(-) Missing the reference mentioned above.
(-) The computational complexity of the method seems to be a limitation. At every task, one needs to carry out the SVD for obtaining M matrix, which could become quite expensive if large network model is used.
(-) In Eq.(4), using sigmoid should be more explicitly explained. I believe the sigmoid is applied at the end of the gradient update, but is \lambda appearing at the right-hand side of (4) also the one with sigmoid, or not? Also, this step seems to be a heuristic, and it's not clear whether this is the best option.
(-) Experiments on larger datasets would show more compelling results.

Overall, despite some weakness mentioned above, I am leaning toward weak accept, but I will also see what other reviewers think about the paper.

-------

I have read the author's rebuttal and other reviewers' reviews, and I would like to keep my score to 6.

**Time Spent Reviewing:**

1.5 hour

---

> ### Author Response · Authors · 2021-08-06
> **Author's Response to Reviewer tT1e**
>
> We’d like to thank the reviewer for taking the time to read our submission and for his/her comments. We address the reviewer’s comments below.
>
> **Reviewer’s Comment 1:** “Missing the reference [1] mentioned above.”
> **Response:** Thanks for your reminder. We will add this reference and discuss it in the related works section of our revised manuscript.
>
> **Reviewer’s Comment 2:** “The computational complexity of the method seems to be a limitation. At every task, one needs to carry out the SVD for obtaining M matrix, which could become quite expensive if a large network model is used.”
> **Response:** The computational complexity of SVD is indeed a limitation of our method for large networks. To address this issue, we only perform SVD once for each layer at the end of each task. In particular, for the widely used *conv* neural networks, we can express the *conv* operator as matrix multiplication by reshaping the input tensor and filters into two matrices (see appendix section C.8 and [2]), then the dimension of the basis of the *conv* is $C_i * k * k$, where $C_i$ and $k$ denote the number of input channels of the *conv* layer and the kernel size of the filters respectively. Thus, SVD can be performed with a much smaller time complexity compared with the fully connected layer. To further clarify the time complexity, we empirically report the training time for each task of the 20-Split CIFAR-100 Superclass on the LeNet-5 architecture with 20-50-800-500 neurons in Figure 5(b) of the main paper. We find that the training time of our method is acceptable. Moreover, we can use off-the-shelf approximation methods [3] to accelerate SVD, which can alleviate the time-consuming problem under large network models.
>
> **Reviewer’s Comment 3:** “In Eq.(4), using sigmoid should be more explicitly explained. I believe the sigmoid is applied at the end of the gradient update, but is \lambda appearing at the right-hand side of (4) also the one with sigmoid, or not? Also, this step seems to be a heuristic, and it's not clear whether this is the best option.”
> **Response:** We thank the reviewer for this comment, which gives us an opportunity to explain clearly. In Eq.(4), we only use the sigmoid at the end of the gradient update. The purpose of the sigmoid function used here is to project the updated value of $\lambda$ (\lambda) to the feasible region $[0,1]$. If $\lambda$ is 0, it means that the corresponding basis is not important to old tasks and can be completely released for learning new tasks. Conversely, if $\lambda$ is 1, the basis is regarded as important to old tasks, and any update in this direction shall be frozen. Beyond this region, any value will be meaningless. We acknowledge that there may be better options than the sigmoid function, and this topic is left for our future exploration.
>
> **Reviewer’s Comment 4:** “Experiments on larger datasets would show more compelling results.”
> **Response:** We agree that it will be more convincing to verify on larger datasets. Thus, in our original submission, we have reported experimental results on the TinyImageNet dataset, which contains 100K images from 200 classes and is much larger than CIFAR100 and miniImageNet datasets used in [2].
>
> [1] *Cha. et al., CPR: Classifier-projection regularization for continual learning, ICLR 2021.*
> [2] *Saha G, Garg I, Roy K. Gradient projection memory for continual learning, ICLR 2021.*
> [3] *Halko N, Martinsson PG, Tropp JA. Finding structure with randomness: Probabilistic algorithms for constructing approximate matrix decompositions. SIAM review. 2011;53(2):217-88.*

---

### Official Review · Reviewer_zbwu · 2021-07-16

**Rating:** 7
**Confidence:** 4

**Summary:**

The article proposes a continual learning method that innovatively improves on an existing algorithm. Precisely, the current work takes GPM (Gradient Projection Memory) which they show is good at preserving performance on old tasks (good stability), but it's not that efficient at learning new ones (low sensitivity) and modifies it in two ways: by dynamically adjusting the importance of the various bases such that the less important ones can be dropped to increase sensitivity. In addition it improves the stability by employing a form of adversarial parameter perturbation to flatten the loss landscape.
The contributions of the paper are:
- an analysis that shows the loss landscape and its "flattness" / "sharpness"; this analysis is used to support the idea that flatter minima tend to forget less in continual learning setups;
- changing the GPM such that it has an increased sensitivity while also improving the stability by flattening the loss landscape around the current solution


**Ethical Concerns:**

No.

**Limitations And Societal Impact:**

Limitations: not really. The approach is applicable to a subset of continual learning scenarios with assumptions such as known task boundaries, control over the times the data is used, etc.
Societal Impact: Not applicable.

**Main Review:**

The current work is novel and relevant to the continual learning literature. It identifies a weakness in a competitive algorithm from the literature (the low "sensitivity" of GPM = low performance on new task) and improves it with a well motivated modification.

The article is well written, the formulas and the pseudocode are clear, and the experiments support the claims made.

The algorithms used for comparison are relevant and diverse. The only weakness of the approach is its limitation to the "known task boundaries" setup and the lack of large-scale or reinforcement learning experiments. Although they are not the most challenging, the set of tasks are adequate and common in the continual learning literature.

Given the originality of the analysis of the sharpness of the loss landscape, the well-motivated changes to GPM, and the empirical results, I consider the paper should be accepted.

Minor observation: Humans are animals.


**Time Spent Reviewing:**

4

---

> ### Author Response · Authors · 2021-08-06
> **Author's Response to Reviewer zbwu**
>
> We thank the reviewer for the review and comments on our paper. We are happy to hear that he/she finds our idea interesting. We address the reviewer’s comment below:
>
> **Reviewer’s Comment:** "The only weakness of the approach is its limitation to the "known task boundaries" setup and the lack of large-scale or reinforcement learning experiments."
>
> **Response:**
> 1. Thanks for your comments. Our work focuses on the "known task boundaries" setup, but it can be easily extended to the "task-agnostic" setting by adding a task-recognition module. For example, we can use the **forget me not process** [1] to recognize tasks like EWC [2].
> 2. We agree that adopting continual learning algorithms for reinforcement learning tasks is valuable, and notice that Wołczyk M et al.[3] recently released a benchmark named **continual world** to encourage the development of continual reinforcement learning. [3] shows that it is challenging and nontrivial to adopt continual learning for reinforcement learning. Thus, continual reinforcement learning remains as an exciting topic for us to be explored in the future.
>
> [1] *Milan K, Veness J, Kirkpatrick J, et al. The forget-me-not process[J]. Advances in Neural Information Processing Systems, 2016, 29: 3702-3710.*
> [2] *Kirkpatrick J, Pascanu R, Rabinowitz N, et al. Overcoming catastrophic forgetting in neural networks[J]. Proceedings of the national academy of sciences, 2017, 114(13): 3521-3526.*
> [3] *Wołczyk M, Zając M, Pascanu R, et al. Continual World: A Robotic Benchmark For Continual Reinforcement Learning[J]. arXiv preprint arXiv:2105.10919, 2021.*

---

### Author Response · Authors · 2021-08-06
**To all reviewers and chairs**

We thank all reviewers for their positive reviews and valuable comments on our paper, and we are happy to hear that all reviewers recognize our work as interesting. We have addressed all concerns and look forward to your feedback!

---

> ### Comment · Area_Chair_avkn · 2021-08-24
> **Discussion**
>
> Can reviewers comment whether the rebuttal change their mind about the paper or answer all of their concerns?
>
> Thank you

---

> > ### Comment · Reviewer_zbwu · 2021-08-30
> > **Discussion**
> >
> > Hi all,
> >
> > I read all the reviews together with the author's responses. My concerns were minor and authors addressed them. Reviewer sRK3 raised some interesting points regarding the flatness of the multi-task solution, and the algorithm's performance for various memory budgets, and the authors provided numbers for that. That only adds to paper's quality, therefore I stand by my initial suggestion for the paper to be accepted.

---

> > ### Comment · Reviewer_hs57 · 2021-09-01
> > **Discussion**
> >
> > After reading the rebuttal and the reviews from other reviewers, my concerns are partially addressed. The idea of exploiting the flatness of loss landscape for continual learning is new and interesting. So I am inclined to keep my score.
> >
> >
> > Best,
> > Yunhui

---

> > ### Comment · Reviewer_sRK3 · 2021-09-03
> > **Re: Discussion**
> >
> > I believe the paper makes important contribution and the authors have addressed my comments/questions during the rebuttal period. So I keep my score.

---

### Decision · Program_Chairs · 2021-09-27

**Decision:**

Accept (Poster)

**Comment:**

Echoing what the reviewers highlighted as well. I think the idea of this work is quite novel, and the paper is well executed, where there was due diligence in terms of explaining the hypothesis, and doing ablation studies. As it stands, I think the work is definitely interesting for the community and should be accepted.